# A pre-registered short-term forecasting study of COVID-19 in Germany and Poland during the second wave

J. Bracher [1,2✉], D. Wolffram [1,2], J. Deuschel [1], K. Görgen [1], J. L. Ketterer [1], A. Ullrich [3], S. Abbott[4], M. V. Barbarossa[5], D. Bertsimas[6], S. Bhatia [7], M. Bodych [8], N. I. Bosse [4], J. P. Burgard [9], L. Castro [10], G. Fairchild [10], J. Fuhrmann[5,11], S. Funk [4], K. Gogolewski [12], Q. Gu [13], S. Heyder [14], T. Hotz[14], Y. Kheifetz[15], H. Kirsten [15], T. Krueger[8], E. Krymova [16], M. L. Li [17], J. H. Meinke [11], I. J. Michaud [18], K. Niedzielewski [19], T. Ożański [8], F. Rakowski[19], M. Scholz [15], S. Soni [6], A. Srivastava [20], J. Zieliński [19], D. Zou[13], T. Gneiting[2,21], M. Schienle [1✉] & List of Contributors by Team*

Disease modelling has had considerable policy impact during the ongoing COVID-19 pandemic, and it is increasingly acknowledged that combining multiple models can improve the reliability of outputs. Here we report insights from ten weeks of collaborative short-term forecasting of COVID-19 in Germany and Poland (12 October–19 December 2020). The study period covers the onset of the second wave in both countries, with tightening non-pharmaceutical interventions (NPIs) and subsequently a decay (Poland) or plateau and renewed increase (Germany) in reported cases. Thirteen independent teams provided probabilistic real-time forecasts of COVID-19 cases and deaths. These were reported for lead times of one to four weeks, with evaluation focused on one- and two-week horizons, which are less affected by changing NPIs. Heterogeneity between forecasts was considerable both in terms of point predictions and forecast spread. Ensemble forecasts showed good relative performance, in particular in terms of coverage, but did not clearly dominate single-model predictions. The study was preregistered and will be followed up in future phases of the pandemic.

[1] Chair of Statistics and Econometrics, Karlsruhe Institute of Technology (KIT), Karlsruhe, Germany. [2] Computational Statistics Group, Heidelberg Institute for Theoretical Studies (HITS), Heidelberg, Germany. [3] Robert Koch Institute (RKI), Berlin, Germany. [4] London School of Hygiene and Tropical Medicine, London, UK. [5] Frankfurt Institute for Advanced Studies, Frankfurt, Germany. [6] Sloan School of Management, Massachusetts Institute of Technology, Cambridge, MA, USA. [7] MRC Centre for Global Infectious Disease Analysis, Abdul Latif Jameel Institute for Disease and Emergency Analytics (J-IDEA), Imperial College London, London, UK. [8] Wroclaw University of Science and Technology, Wroclaw, Poland. [9] Economic and Social Statistics Department, University of Trier, Trier, Germany. [10] Information Systems and Modeling, Los Alamos National Laboratory, Los Alamos, NM, USA. [11] Jülich Supercomputing Centre, Forschungszentrum Jülich, Jülich, Germany. [12] Institute of Informatics, University of Warsaw, Warsaw, Poland. [13] Department of Computer Science, University of California, Los Angeles, CA, USA. [14] Institute of Mathematics, Technische Universität Ilmenau, Ilmenau, Germany. [15] Institute for Medical Informatics, Statistics and Epidemiology, University of Leipzig, Leipzig, Germany. [16] Swiss Data Science Center, ETH Zurich and EPFL, Lausanne, Switzerland. [17] Operations Research Center, Massachusetts Institute of Technology, Cambridge, MA, USA. [18] Statistical Sciences Group, Los Alamos National Laboratory, Los Alamos, NM, USA. [19] Interdisciplinary Centre for Mathematical and Computational Modeling, University of Warsaw, Warsaw, Poland. [20] Ming Hsieh Department of Computer and Electrical Engineering, University of Southern California, Los Angeles, CA, USA. [21] Institute for Stochastics, Karlsruhe Institute of Technology (KIT), Karlsruhe, Germany. *A list of authors and their affiliations appears at the end of the paper. ✉email: johannes.bracher@kit.edu; melanie.schienle@kit.edu

Forecasting is one of the key purposes of epidemic modelling, and despite being related to the understanding of underlying mechanisms, it is a conceptually distinct task[1,2]. Explanatory models are often strongly idealised and tailored to specific settings, aiming to shed light on latent biological or social mechanisms. Forecast models, on the other hand, have a strong focus on observable quantities, aiming for quantitatively accurate predictions in a wide range of situations. While understanding of mechanisms can provide guidance to this end, forecast models may also be purely data-driven. Accurate disease forecasts can improve situational awareness of decision makers and facilitate tasks such as resource allocation or planning of vaccine trials[3]. During the COVID-19 pandemic, there has been a surge in research activity on epidemic forecasting. Contributions vary greatly in terms of purpose, forecast targets, methods, and evaluation criteria. An important distinction is between longer-term scenario or what-if projections and short-term forecasts[4]. The former attempt to discern the consequences of hypothetical scenarios (e.g., intervention strategies), a task closely linked to causal statements as made by explanatory models. Scenarios typically remain counterfactuals and thus cannot be evaluated directly using subsequently observed data. Short-term forecasts, on the other hand, refer to brief time horizons, at which the predicted quantities are expected to be largely unaffected by yet unknown changes in public health interventions. This makes them particularly suitable to assess the predictive power of computational models, a need repeatedly expressed during the pandemic[5].

Rigorous assessment of forecasting methods should follow several key principles. Firstly, forecasts should be made in real time, as retrospective forecasting often leads to overly optimistic conclusions about performance. Real-time forecasting poses many challenges[6], including noisy or delayed data, incomplete knowledge on testing and interventions as well as time pressure. Even if these are mimicked in retrospective studies, some benefit of hindsight remains. Secondly, in a pandemic situation with low predictability, forecast uncertainty needs to be quantified explicitly[7,8]. Lastly, forecast studies are most informative if they involve comparisons between multiple independently run methods[9]. Such collaborative efforts have led to important advances in short-term disease forecasting prior to the pandemic[10–13]. Notably, they have provided evidence that ensemble forecasts combining various independent predictions can lead to improved performance, similar to what has been observed in weather prediction[14].

The German and Polish COVID-19 Forecast Hub is a collaborative project which, guided by the above principles, aims to collect, evaluate and combine forecasts of weekly COVID-19 cases and deaths in the two countries. It is run in close exchange with the US COVID-19 Forecast Hub[15,16] and aims for compatibility with the forecasts assembled there. Close links moreover exist to a similar effort in the United Kingdom[17]. Other conceptually related works on short-term forecasting or baseline projections include those by consortia from Austria[18] and Australia[19] as well as the European Centre for Disease Prevention and Control[20,21] (ECDC). In a German context, various nowcasting efforts exist[22]. All forecasts assembled in the German and Polish COVID-19 Forecast Hub are publicly available (https://github.com/KITmetricslab/covid19-forecast-hub-de[23]) and can be explored interactively in a dashboard (https://kitmetricslab.github.io/forecasthub). The Forecast Hub project moreover aims to foster exchange between research teams from Germany, Poland and beyond. To this end, regular video conferences with presentations on forecast methodologies, discussions and feedback on performance were organised.

In this work, we present results from a prospective evaluation study based on the collected forecasts. The evaluation procedure was prespecified in a study protocol[24], which we deposited at the registry of the Open Science Foundation (OSF) on 8 October 2020. The evaluation period extends from 12 October 2020 (first forecasts issued) to 19 December 2020 (last observations made). This corresponds to the onset of the second wave of the pandemic in both countries. It is marked by strong virus circulation and changes in intervention measures and testing strategies, see Fig. 1 for an overview. This makes for a situation in which reliable short-term forecasting is both particularly useful and particularly challenging. Thirteen modelling teams from Germany, Poland, Switzerland, the United Kingdom and the United States contributed forecasts of weekly confirmed cases and deaths. Both targets are addressed on the incidence and cumulative scales and one through 4 weeks ahead, with evaluation focused on 1 and 2 weeks ahead. We find considerable heterogeneity between forecasts from different models and an overall tendency to overconfident forecasting, i.e., lower than nominal coverage of prediction intervals. While for deaths, a number of models were able to outperform a simple baseline forecast up to 4 weeks into the future, such improvements were limited to shorter horizons for cases. Combined ensemble predictions show good relative performance in particular in terms of interval coverage, but do not clearly dominate single-model predictions. Conclusions from 10 weeks of real-time forecasting are necessarily preliminary, but we hope to contribute to an ongoing exchange on best practices in the field. Note that the considered period is the last one to be unaffected by vaccination and caused exclusively by the "original" wild type variant of the virus. Early January marked both the start of vaccination campaigns and the likely introduction of the B.1.1.7 (alpha) variant of concern in both countries. Our study will be followed up until at least March 2021 and may be extended beyond.

## Results

In the following we provide specific observations made during the evaluation period as well as a formal statistical assessment of performance. Particular attention is given to combined ensemble forecasts. Forecasts refer to data from the European Centre for Disease Prevention and Control[25] (ECDC) or Johns Hopkins University Centre for Systems Science and Engineering[26] (JHU CSSE); see the Methods section for the exact definition of targets and ensemble methods. Visualisations of 1- and 2-week-ahead forecasts on the incidence scale are displayed in Figs. 2 and 3, respectively, and will be discussed in the following. These figures are restricted to models submitted over (almost) the entire evaluation period and providing complete forecasts with 23 predictive quantiles. Forecasts from the remaining models are illustrated in Supplementary Note 7. Forecasts at prediction horizons of 3 and 4 weeks are shown in Supplementary Note 8. All analyses of forecast performance were conducted using the R language for statistical computing[27].

**Heterogeneity between forecasts**. A recurring theme during the evaluation period was pronounced variability between model forecasts. Figure 4 illustrates this aspect for point forecasts of incident cases in Germany. The left panel shows the spread of point forecasts issued on 19 October 2020 and valid 1 to 4 weeks ahead. The models present very different outlooks, ranging from a return to the lower incidence of previous weeks to exponential growth. The graph also illustrates the difficulty of forecasting cases >2 weeks ahead. Several models had correctly picked up the upwards trend, but presumably a combination of the new testing regime and the semi-lockdown (marked as (a) and (b)) led to a flattening of the curve. The right panel shows forecasts from 9 November 2020, immediately following the aforementioned

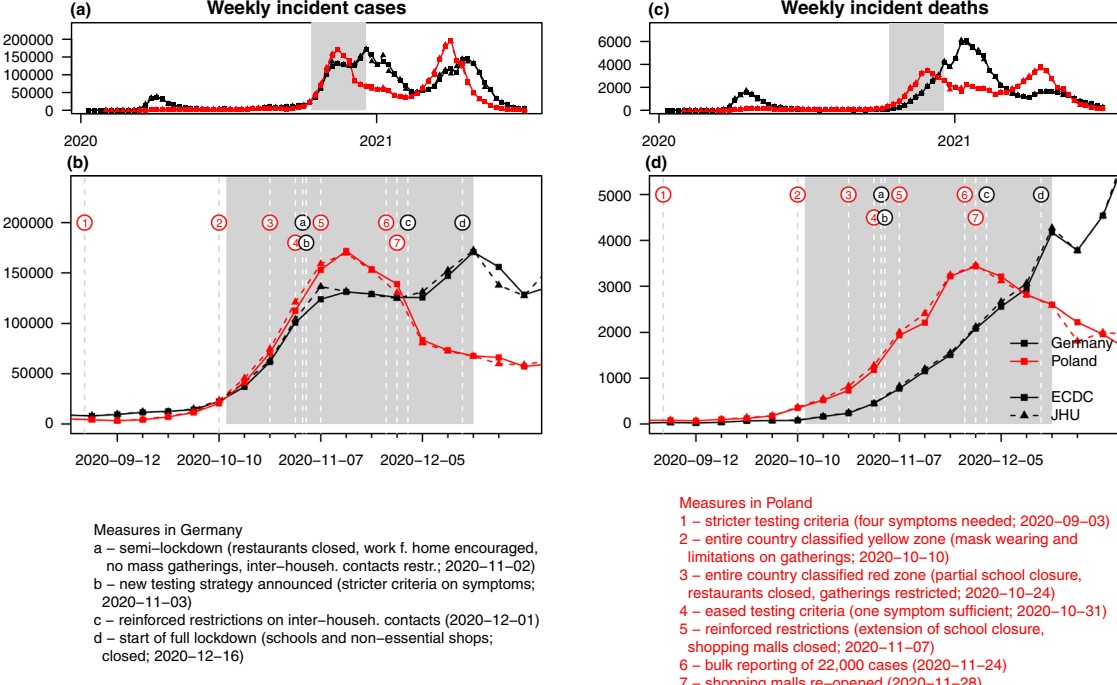

**Fig. 1 Forecast evaluation period.** Weekly incident (**a**, **b**) confirmed cases and (**c**, **d**) deaths from COVID-19 in Germany and Poland according to data sets from the European Centre for Disease Prevention and Control (ECDC) and the Centre for Systems Science and Engineering at Johns Hopkins University (JHU). The study period covered in this paper is highlighted in grey. Important changes in interventions and testing are marked by letters/numbers and dashed vertical lines. Sources containing details on the listed interventions are provided in Supplementary Note 5.

events. Again, the forecasts are quite heterogeneous. The week ending on Saturday 7 November had seen a slower increase in reported cases than anticipated by almost all models (see Fig. 2), but there was general uncertainty about the role of saturating testing capacities and evolving testing strategies. Indeed, on 18 November it was argued in a situation report from Robert Koch Institute (RKI) that comparability of data from calendar week 46 (9–15 November) to previous weeks is limited[28]. This illustrates that confirmed cases can be a moving target, and that different modelling decisions can lead to very different forecasts.

Forecasts are not only heterogeneous with respect to their central tendency, but also the implied uncertainty. As can be seen from Figs. 2 and 3, certain models issue very confident forecasts with narrow forecast intervals barely visible in the plot. Others— in particular LANL-GrowthRate and the exponential smoothing time series model KIT-time_series_baseline— show rather large uncertainty. For almost all forecast dates there are pairs of models with no or minimal overlap in 95% prediction intervals, another indicator of limited agreement between forecasts. As can be seen from the right column of Figs. 2 and 3 as well as Tables 1 and 2, most contributed models were overconfident, i.e., their prediction intervals did not reach nominal coverage.

**Adaptation to changing trends and truth data issues.** Far from all forecast models explicitly account for interventions and testing strategies (Table 3). Many forecasters instead prefer to let their models pick up trends from the data once they become apparent. This can lead to delayed adaptation to changes and explains why numerous models—including the ensemble—showed overshoot in the first half of November when cases started to plateau in Germany (visible from Fig. 2 and even more pronounced in Fig. 3). Interestingly, some models adapted more quickly to the flatter curve. This includes the human judgement approach

EpiExpert, which, due to its reliance on human input, can take information on interventions into account before they become apparent in epidemiological data, but interestingly also Epi1Ger and EpiNow2 which do not account for interventions. In Poland, overshoot could be observed following the peak week in cases (ending on 15 November), with the 1-week-ahead median ensemble only barely covering the next observed value. However, most models adapted quickly and were back on track in the following week.

A noteworthy difficulty for death forecasts in Germany was under-prediction in consecutive weeks in late November and December. In November, several models predicted that death numbers would level off, likely as a consequence of the plateau in case numbers starting several weeks before. In the last week of our study (ending on 19 December) most models considerably under-estimated the increase in weekly deaths. A difficulty may have been that despite the overall plateau observed until early December, cases continued to increase in the oldest age groups, for which the mortality risk is highest (Supplementary Fig. 1). Models that ignore the age structure of cases— which includes most available models (Table 3)— may then have been led astray.

A major question in epidemic modelling is how closely surveillance data reflect the underlying dynamics. Like in Germany, testing criteria were repeatedly adapted in Poland. In early September they were tightened, requiring the simultaneous presence of four symptoms for the administration of a test. This was changed to less restrictive criteria in late October (presence of a single symptom). These changes limit comparability of numbers across time. Very high test positivity rates in Poland (Supplementary Fig. 2) suggest that there was substantial under-ascertainment, which is assumed to have aggravated over time. Comparisons between overall excess mortality and reported COVID deaths suggest that there is also relevant under-ascertainment of deaths, again likely changing over time[29]. These aspects make predictions challenging, and limitations of ground

truth data sources are inherited by the forecasts which refer to them. A striking example of this was the belated addition of 22,000 cases from previous weeks to the Polish record on 24 November 2020. The Poland-based teams MOCOS and MIMUW explicitly took this shift into account while other teams did not.

**Findings for median, mean and inverse-WIS ensembles**. We assessed the performance of forecast ensembles based on various aggregation rules, more specifically a median, a mean and an inverse-WIS (weighted interval score) ensemble; see the Methods section for the respective definitions.

A key advantage of the median ensemble is that it is more robust to single extreme forecasts than the mean ensemble. As an example of the behaviour when one forecast differs considerably from the others we show forecasts of incident deaths in Poland from 30 November 2020 in Fig. 5. The first panel shows the six member forecasts, the second the resulting median and mean ensembles. The predictive median of the latter is noticeably higher as it is more strongly impacted by one model which predicted a resurge in deaths.

A downside of the median ensemble is that its forecasts are not always well-shaped, in particular when a small to medium number of heterogeneous member forecasts is combined. A pronounced example is shown in the third and fourth panel of Fig. 5. For the 1-week-ahead forecast of incident cases in Poland from 2 November 2020, the predictive 25% quantile and median were almost identical. For the 2-week-ahead median ensemble forecast, the 50% and 75% quantile were almost identical. Both distributions are thus rather oddly shaped, with a quarter of the probability mass concentrated in a short interval. The mean ensemble, on the other hand, produces a more symmetric and thus more realistic representation of the associated uncertainty.

We briefly address the inverse-WIS ensemble, which is a pragmatic approach to giving more weight to forecasts with good recent performance. Figure 6 shows the weights of the various member models for incident deaths in Germany and Poland. Note that some models were not included in the ensemble in certain weeks, either because of delayed or missing submissions or due to concerns about their plausibility. While certain models on average receive larger weights than others, weights change considerably over time. These fluctuations make it challenging to improve ensemble forecasts by taking past performance into account, and indeed Tables 1 and 2 do not indicate any systematic benefits from inverse-WIS weighting. A possible reason is that models get updated continuously by their maintainers, including major revisions of methodology.

**Formal forecast evaluation**. Forecasts were evaluated using the mean-weighted interval score (WIS), mean absolute error (AE) and interval coverage rates. The WIS is a generalisation of the absolute error to probabilistic forecasts and negatively oriented, meaning that smaller values are better (see the Methods section). Tables 1 and 2 provide a detailed overview of results by country, target and forecast horizon, based on data from the European Centre for Disease Prevention and Control[25] (ECDC). We repeated all evaluations using data from the Centre for Systems Science and Engineering at Johns Hopkins University[26] (JHU CSSE) as ground truth (Supplementary Note 7), and the overall results seem robust to this choice. We also report on 3- and 4-week-ahead forecasts in Supplementary Note 8, though for reasons discussed in the Methods section, we consider their usability limited. To put the results of the submitted and ensemble forecasts into perspective we created forecasts from three baseline methods of varying complexity, see Methods section.

Figure 7 depicts the mean WIS achieved by the different models on the incidence scale. For models providing only point forecasts, the mean AE is shown, which as detailed in the Methods section, can be compared to mean WIS values. A simple model always predicting the same number of new cases/deaths as in the past week (KIT-baseline) serves as a reference. For deaths, the ensemble forecasts and several submitted models outperform this baseline up to three or even 4 weeks ahead. Deaths are a more strongly lagged indicator, which favours predictability at somewhat longer horizons. Another aspect may be that at least in Germany, death numbers have been following a rather uniform upward trend over the study period, making it relatively easy to beat the baseline model. For cases, which are a more immediate measure, almost none of the compared approaches meaningfully outperformed the naive baseline beyond a horizon of 1 or 2 weeks. Especially in Germany this result is largely due to the aforementioned overshoot of forecasts in early November. The KIT-baseline forecast always predicts a plateau, which is what was observed in Germany for roughly half of the evaluation period. Good performance of the baseline is thus less surprising. Nonetheless, these results underscore that in periods of evolving intervention measures meaningful case forecasts are limited to a rather short time window. In this context we also note that the additional baselines KIT-extrapolation_baseline and KIT-time_series_baseline do not systematically outperform the naive baseline and for most targets are neither among the best nor the worst performing approaches.

In exploratory analyses (Supplementary Fig. 9) we did not find any clear indication that certain modelling strategies (defined via the five categories used in Table 3) performed better than others. Following changes in trends, the human judgement model epiforecasts-EpiExpert showed good average performance, while growth rate approaches had a stronger tendency to overshoot (Supplementary Figs. 5–8). Otherwise, variability of performance within model categories was pronounced and no apparent patterns emerged.

The median, mean and inverse-WIS ensembles showed overall good, but not outstanding relative performance in terms of mean WIS. At a 1-week lead time, the median ensemble outperformed the baseline forecasts quite consistently for all considered targets, showing less variable performance than most member models (Supplementary Figs. 5–8). Differences between the ensemble approaches are minor and do not indicate a clear ordering. We re-ran the ensembles retrospectively using all available forecasts, i.e., including those submitted late or excluded due to implausibilities. As can be seen from Supplementary Tables 5 and 6, this led only to minor changes in performance. Unlike in the US effort[30,31], the ensemble forecast is not strictly better than the single-model forecasts. Typically, performance is similar to some of the better-performing contributed forecasts, and sometimes the latter have a slight edge (e.g., FIAS_FZJ-Epi1Ger for cases in Germany and MOCOS-agent1 for deaths in Poland). Interestingly, the expert forecast epiforecasts-EpiExpert is often among the more successful methods, indicating that an informed human assessment sets a high bar for more formalised model-based approaches. In terms of point forecasts, the extrapolation approach SDSC_ISG-TrendModel shows good relative performance, but only covers 1-week-ahead forecasts.

The 50% and 95% prediction intervals of most forecasts did not achieve their respective nominal coverage levels (most apparent for cases 2 weeks ahead). The statistical time series model KIT-time_series_baseline features favourably here, though at the expense of wide forecast intervals (Fig. 2). While its lack of sharpness leads to mediocre overall performance in terms of the WIS, the model seems to have been a helpful addition to the ensemble by counterbalancing the overconfidence of other

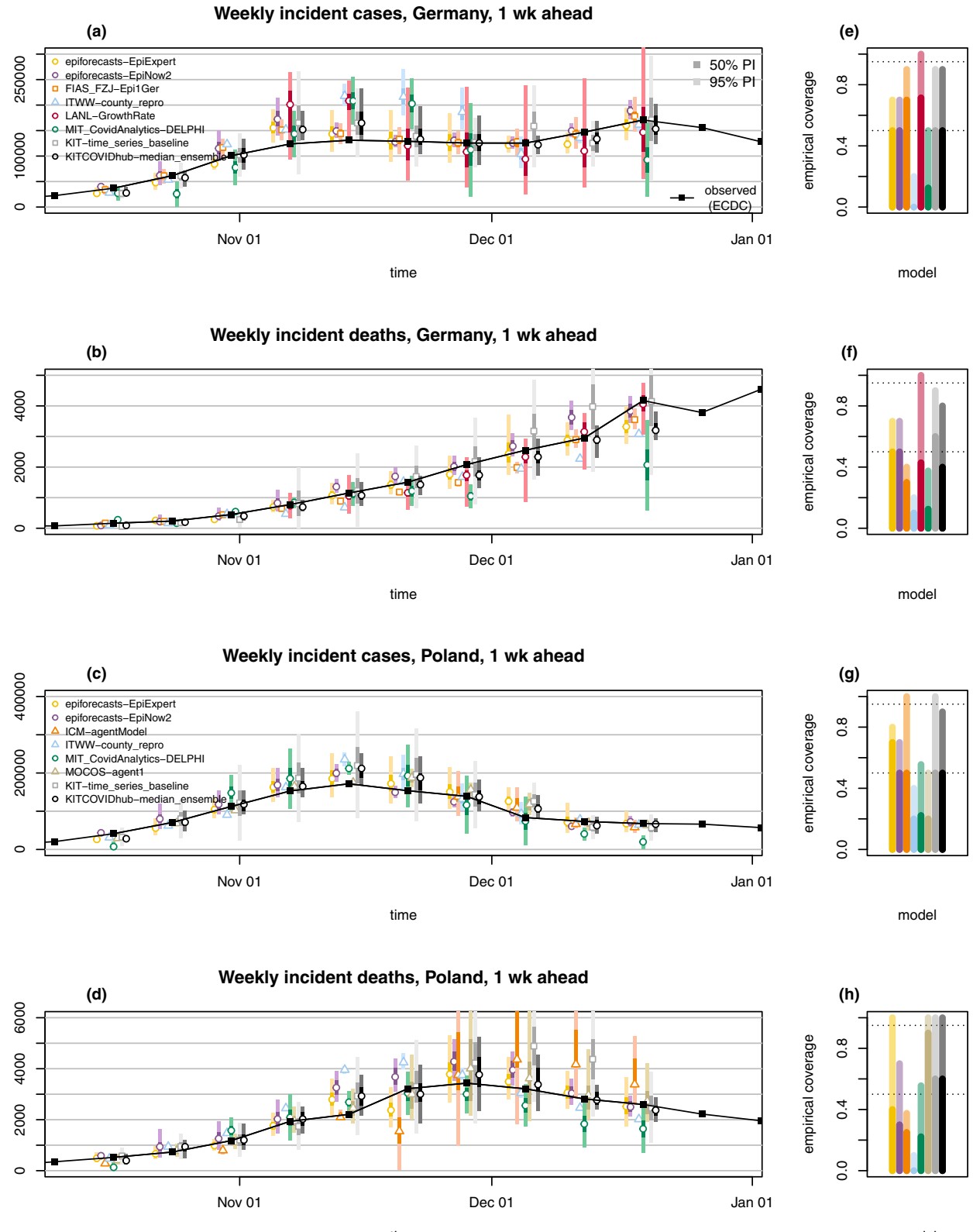

**Fig. 2 One-week-ahead forecasts.** One-week-ahead forecasts of incident cases and deaths in Germany (**a**, **b**) and Poland (**c**, **d**). Displayed are predictive medians, 50% and 95% prediction intervals (PIs). Coverage plots (**e–h**) show the empirical coverage of 95% (light) and 50% (dark) prediction intervals.

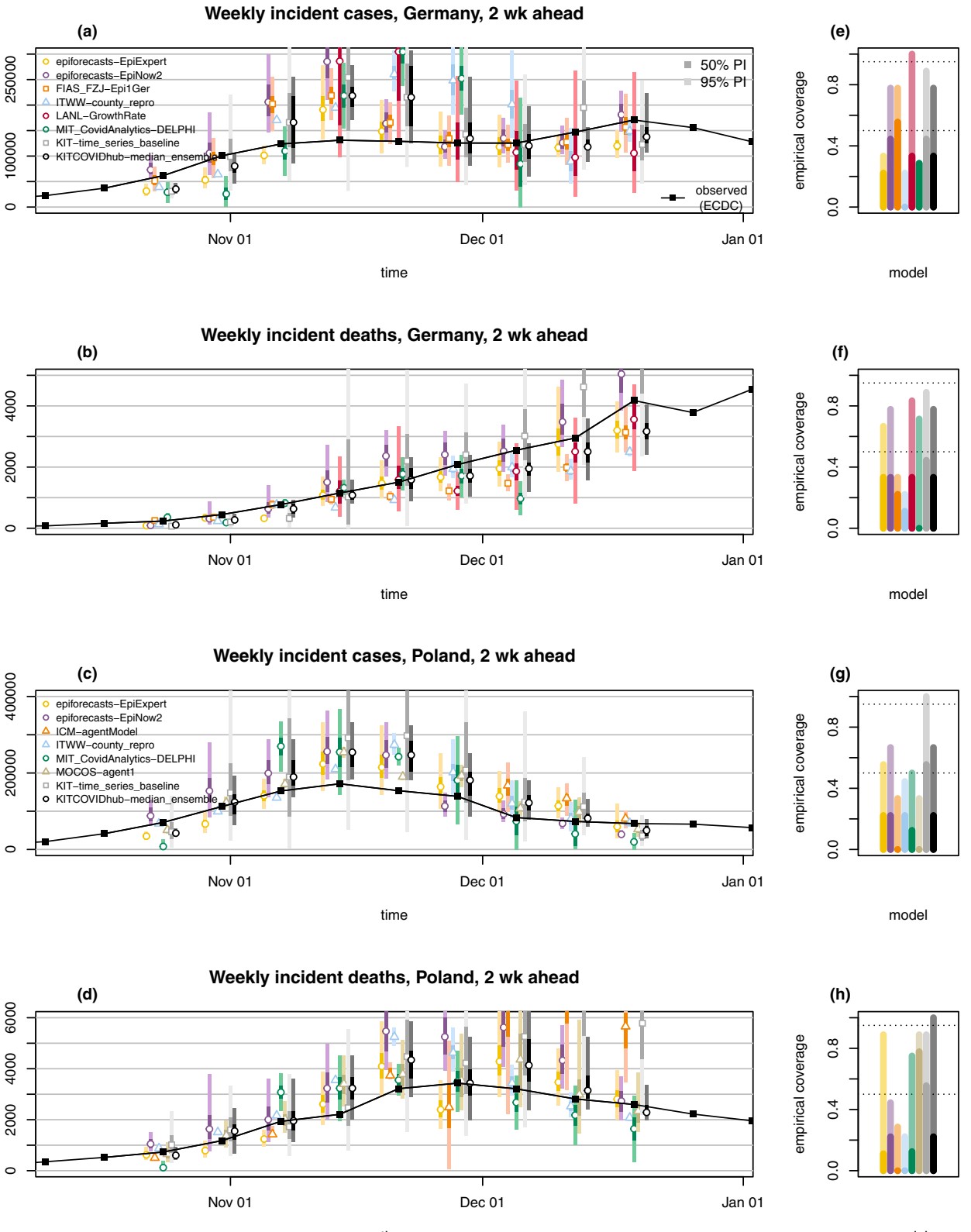

**Fig. 3 Two-week-ahead forecasts.** Two-week-ahead forecasts of incident cases and deaths in Germany (**a**, **b**) and Poland (**c**, **d**). Displayed are predictive medians, 50% and 95% prediction intervals (PIs). Coverage plots (**e–h**) show the empirical coverage of 95% (light) and 50% (dark) prediction intervals.

models. Indeed, coverage of the 95% intervals of the ensemble is above average, despite not reaching nominal levels.

A last aspect worth mentioning concerns the discrepancies between results for 1-week-ahead incident and cumulative

quantities. In principle these two should be identical, as forecasts should only be shifted by an additive constant (the last observed cumulative number). This, however, was not the case for all submitted forecasts, and coherence was not enforced by our

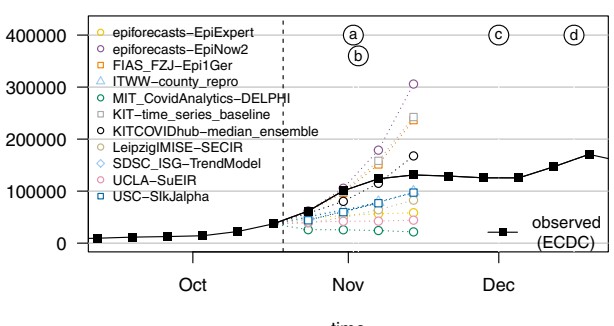

**(a) Incident cases, Germany, forecasts from 2020–10–19**

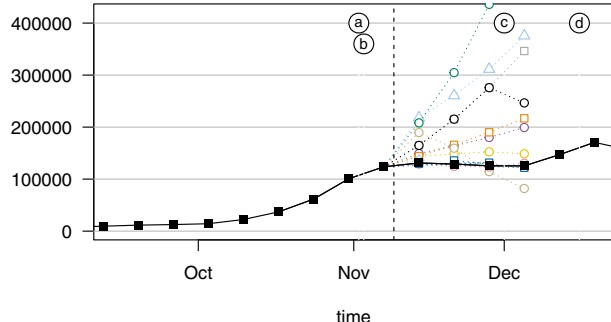

**(b) Incident cases, Germany, forecasts from 2020–11–09**

**Fig. 4 Illustration of heterogeneity between incident case forecasts in Germany. a** Point forecasts issued by different models and the median ensemble on 19 October 2020. **b** Point forecasts issued on 9 November 2020. The dashed vertical line indicates the date at which forecasts were issued. Events marked by letters a–d are explained in Fig. 1.

submission system. For the ensemble forecasts the discrepancies are largely due to the fact that the included models are not always the same.

## Discussion
We presented results from a preregistered forecasting project in Germany and Poland, covering 10 weeks during the second wave of the COVID-19 pandemic. We believe that such an effort is helpful to put the outputs from single models in context, and to give a more complete picture of the associated uncertainties. For modelling teams, short-term forecasts can provide a useful feedback loop, via a set of comparable outputs from other models, and regular independent evaluation. A substantial strength of our study is that it took place in the framework of a prespecified evaluation protocol. The criteria for evaluation were communicated in advance, and most considered models covered the entire study period.

Similarly to Funk et al.[17], we conclude that achieving good predictive accuracy and calibration is challenging in a dynamic epidemic situation. Epidemic forecasting is complicated by numerous challenges absent in, e.g., weather forecasting[32]. Noisy and delayed data are an obstacle, but the more fundamental difficulty lies in the complex social (and political) dynamics shaping an epidemic[33]. These are more relevant for major outbreaks of emerging diseases than for seasonal diseases, and limit predictability to rather short time horizons.

Not all included models were designed for the sole purpose of short-term forecasting, and could be tailored more specifically to this task. Certain models were originally conceived for what-if projections and retrospective assessments of longer-term dynamics and interventions. This focus on a global fit may limit their flexibility to align closely with the most recent data, making them less successful at short forecast horizons compared to simpler extrapolation approaches. We observed pronounced heterogeneity between the different forecasts, with a general tendency to overconfident forecasting. While over the course of 10 weeks, some models achieved better average scores than others, relative performance has been fluctuating considerably.

Various works on multi-model disease forecasting discuss performance differences between modelling approaches, most commonly between mechanistic and statistical approaches. Reich et al.[13], McGowan et al.[34] (both seasonal influenza) and Johansson et al.[12] (dengue) find slightly better performance of statistical than mechanistic models. All these papers find ensemble approaches to perform best. Forecasting of seasonal and emerging diseases, however, differ in important ways, the latter typically being subject to more variation in reporting procedures

and interventions. This, along with the limited amount of historical data, may benefit mechanistic models. In our study we did not find any striking patterns, but this may be due to the relatively short study period. We expect that forecast performance is also shaped by numerous other factors, including methods used for model calibration, the thoroughness of manual tuning and input on new intervention measures or population behaviour.

Different models may be particularly suitable for different phases of an epidemic[17], which is exemplified by the fact that some models were quicker to adjust to slowing growth of cases in Germany. In particular, we noticed that forecasts based on human assessment performed favourably immediately after changes in trends. These aspects highlight the importance of considering several independently run models rather than focusing attention on a single one, as is sometimes the case in public discussions. Here, collaborative forecasting projects can provide valuable insights and facilitate communication of results. Overall, ensemble methods showed good, but not outstanding relative performance, notably with clearly above-average coverage rates and more stable performance over time. An important question is whether ensemble forecasts could be improved by sensible weighting of members or post-processing steps. Given the limited amount of available forecast history and rapid changes in the epidemic situation, this is a challenging encounter, and indeed we did not find benefits in the inverse-WIS approach.

An obvious extension to both assess forecasts in more detail and make them more relevant to decision makers is to issue them at a finer geographical resolution. During the evaluation period covered in this work, only three of the contributed forecast models (`ITWW-county_repro` and `USC-SIkJalpha`, `LeipzigIMISE-SECIR` for the state of Saxony) also provided forecasts at the sub-national level (German states, Polish voivodeships). Extending this to a larger number of models is a priority for the further course of the project.

In its present form, the platform covers only forecasts of confirmed cases and deaths. These commonly addressed forecasting targets were already covered by a critical mass of teams when the project was started. Given limited available time resources of teams, a choice was made to focus efforts on this narrow set of targets. The was also motivated by the strong focus German legislators have put on seven-day incidences, which have been the main criteria for the strengthening or alleviation of control measures. However, there is an ongoing debate on the usefulness of this indicator, with frequent claims to replace it by hospital admissions[35]. An extension to this target was considered, but in view of emerging parallel efforts and open questions on data availability not prioritised. Given that in a post-vaccination

**Table 1 Detailed summary of forecast evaluation for Germany (based on ECDC data). $C_{0.5}$ and $C_{0.95}$ denote coverage rates of the 50% and 95% prediction intervals; AE and WIS stand for the mean absolute error and mean weighted interval score.**

**Germany, cases**

| Model | 1-week ahead incident | | | | 2-week ahead incident | | | | 1-week ahead cumulative | | | | 2-week ahead cumulative | | | |
|---|---|---|---|---|---|---|---|---|---|---|---|---|---|---|---|---|
| | AE | WIS | $C_{0.5}$ | $C_{0.95}$ | AE | WIS | $C_{0.5}$ | $C_{0.95}$ | AE | WIS | $C_{0.5}$ | $C_{0.95}$ | AE | WIS | $C_{0.5}$ | $C_{0.95}$ |
| epiforecasts-EpiExpert | 12,333 | 8,781 | 5/10 | 7/10 | 30,329 | 22,157 | 2/9 | 3/9 | 12,334 | 8,781 | 5/10 | 7/10 | 42,667 | 30,669 | 2/9 | 4/9 |
| epiforecasts-EpiNow2 | 11,171 | 7,932 | 5/10 | 7/10 | 37,338 | 27,293 | 4/9 | 7/9 | 11,171 | 7,932 | 5/10 | 7/10 | 47,738 | 34,253 | 4/9 | 6/9 |
| FIAS_FZJ-Epi1Ger | 7,798 | 5,709 | 7/10 | 9/10 | 29,190 | 21,058 | 5/9 | 7/9 | 17,255 | 11,264 | 3/10 | 7/10 | 38,925 | 29,937 | 5/9 | 6/9 |
| ITWW-county_repro | 34,425 | 28,906 | 0/10 | 2/10 | 64,378 | 53,136 | 0/9 | 2/9 | 34,077 | 28,558 | 0/10 | 2/10 | 101,184 | 84,276 | 0/9 | 2/9 |
| LANL-GrowthRate | 38,970[a] | 23,379[a] | 5/7 | 7/7 | 77,438[a] | 42,294[a] | 2/6 | 6/6 | 39,042 | 26,794 | 5/10 | 7/10 | 116,494 | 78,224 | 3/9 | 5/9 |
| LeipzigIMISE-SECIR | 20,019 | | 2/5 | 3/5 | 51,115 | | 0/4 | 1/4 | 35,901 | 31,690 | 1/10 | 1/10 | 93,111 | 83,670 | 1/9 | 2/9 |
| MIT_CovidAnalytics-DELPHI | 41,313[a] | 29,004[a] | 1/8 | 4/8 | 78,872[a] | 61,447[a] | 2/7 | 2/7 | | | | | | | | |
| SDSC_ISG-TrendModel | 10,963 | | | | 47,747 | | | | 10,963 | | | | 69,800 | | | |
| UCLA-SuEIR | 25,012 | | | | 30,891 | | 3/9 | | 25,012 | | | | 49,640 | | | |
| USC-SIkJalpha | 20,028 | | 1/1 | 1/1 | 32,690 | | 3/9 | 6/9 | 21,567 | | 1/1 | 1/1 | 47,472 | 37,155 | 3/9 | 5/9 |
| KIT-baseline | 18,475 | 12,998 | 5/10 | 9/10 | 36,498 | 25,543 | 6/9 | 7/9 | 18,475 | 12,998 | 5/10 | 9/10 | 47,145 | 35,142 | 7/9 | 7/9 |
| KIT-extrapolation_baseline | 12,016 | 10,522 | 7/10 | 10/10 | | 26,195 | | | 12,016 | 10,522 | 7/10 | 10/10 | | | | |
| KIT-time_series_baseline | 15,383 | 11,014 | 5/10 | 9/10 | 44,481 | 28,625 | 4/9 | 8/9 | 15,383 | 11,014 | 5/10 | 9/10 | 61,489 | 39,244 | 4/9 | 8/9 |
| KITCOVIDhub-inverse_wis_ensemble | 14,017 | 9,358 | 5/10 | 9/10 | 42,063 | 27,993 | 2/9 | 5/9 | 13,464 | 9,265 | 6/10 | 9/10 | 52,972 | 35,194 | 2/9 | 8/9 |
| KITCOVIDhub-mean_ensemble | 16,649 | 10,677 | 4/10 | 8/10 | 42,214 | 27,290 | 1/9 | 6/9 | 15,771 | 10,630 | 4/10 | 8/10 | 57,125 | 37,356 | 1/9 | 6/9 |
| KITCOVIDhub-median_ensemble | 11,534 | 8,094 | 5/10 | 9/10 | 37,620 | 25,017 | 3/9 | 7/9 | 12,877 | 9,243 | 6/10 | 7/10 | 49,438 | 34,461 | 2/9 | 6/9 |

**Germany, deaths**

| Model | 1-week ahead incident | | | | 2-week ahead incident | | | | 1-week ahead cumulative | | | | 2-week ahead cumulative | | | |
|---|---|---|---|---|---|---|---|---|---|---|---|---|---|---|---|---|
| | AE | WIS | $C_{0.5}$ | $C_{0.95}$ | AE | WIS | $C_{0.5}$ | $C_{0.95}$ | AE | WIS | $C_{0.5}$ | $C_{0.95}$ | AE | WIS | $C_{0.5}$ | $C_{0.95}$ |
| epiforecasts-EpiExpert | 187 | 131 | 5/10 | 7/10 | 333 | 234 | 3/9 | 6/9 | 187 | 131 | 5/10 | 7/10 | 442 | 295 | 3/9 | 7/9 |
| epiforecasts-EpiNow2 | 180 | 120 | 5/10 | 7/10 | 376 | 235 | 3/9 | 7/9 | 180 | 120 | 5/10 | 7/10 | 539 | 336 | 4/9 | 7/9 |
| FIAS_FZJ-Epi1Ger | 256 | 223 | 3/10 | 4/10 | 525 | 433 | 2/9 | 3/9 | 215 | 175 | 3/10 | 4/10 | 684 | 564 | 1/9 | 4/9 |
| Imperial-ensemble2 | 254 | 195 | 5/10 | 5/10 | 537 | 483 | 1/9 | 2/9 | 252 | 192 | 5/10 | 5/10 | 824 | 757 | 2/9 | 2/9 |
| ITWW-county_repro | 371 | 355 | 1/10 | 2/10 | | | | | 370 | 354 | 1/10 | 2/10 | 560 | 441 | 3/9 | 5/9 |
| LANL-GrowthRate | 195[a] | 128[a] | 3/7 | 7/7 | 457[a] | 313[a] | 2/6 | 5/6 | 189 | 134 | 3/10 | 7/10 | 2,184 | 1,799 | 0/9 | 1/9 |
| LeipzigIMISE-SECIR | 621 | | 0/5 | 1/5 | 768 | | 1/4 | 1/4 | 1,167 | 991 | 0/10 | 1/10 | 639[a] | 525[a] | 0/7 | 3/7 |
| MIT_CovidAnalytics-DELPHI | 474[a] | 357[a] | 1/8 | 3/8 | 403[a] | 306[a] | 0/7 | 5/7 | 449[a] | 331[a] | 1/8 | 3/8 | | | | |
| SDSC_ISG-TrendModel | 357 | | | | 827 | | | | 357 | | | | 1,177 | | | |
| UCLA-SuEIR | 456 | | | | 600 | | | | 456 | | | | 963 | | | |
| USC-SIkJalpha | 489 | | 0/1 | 0/1 | 835 | | 0/9 | 5/9 | 500 | | 0/1 | 0/1 | 1,155 | 707 | 0/9 | 5/9 |
| KIT-baseline | 479 | 263 | 2/10 | 9/10 | 383 | 510 | 5/9 | 8/9 | 479 | 263 | 2/10 | 9/10 | 493 | 330 | 5/9 | 8/9 |
| KIT-extrapolation_baseline | 202 | 134 | 7/10 | 9/10 | 624 | 246 | 4/9 | 8/9 | 202 | 134 | 7/10 | 9/10 | 866 | 577 | 4/9 | 8/9 |
| KIT-time_series_baseline | 238 | 190 | 6/10 | 9/10 | 255 | 415 | 2/9 | 8/9 | 238 | 190 | 6/10 | 9/10 | 370 | 224 | 3/9 | 8/9 |
| KITCOVIDhub-inverse_wis_ensemble | 180 | 114 | 4/10 | 9/10 | 298 | 147 | 2/9 | 8/9 | 174 | 108 | 5/10 | 5/10 | 441 | 262 | 2/9 | 8/9 |
| KITCOVIDhub-mean_ensemble | 204 | 138 | 3/10 | 9/10 | 334 | 174 | 3/9 | 7/9 | 216 | 147 | 3/10 | 9/10 | 441 | 262 | 2/9 | 8/9 |
| KITCOVIDhub-median_ensemble | 200 | 135 | 4/10 | 8/10 | 334 | 216 | 3/9 | 7/9 | 202 | 133 | 4/10 | 8/10 | 440 | 270 | 3/9 | 8/9 |

[a]Entries where scores were imputed for at least 1 week. Weighted interval scores and absolute errors were imputed with the worst (largest) score achieved by any other forecast for the respective target and week. Models marked thus received a pessimistic assessment of their performance. If a model covered less than two-thirds of the evaluation period, results are omitted.

**Table 2 Detailed summary of forecast evaluation for Poland (based on ECDC data). $C_{0.5}$ and $C_{0.95}$ denote coverage rates of the 50% and 95% prediction intervals; AE and WIS stand for the mean absolute error and mean-weighted interval score.**

**Poland, cases**

| Model | 1-week ahead incident AE | WIS | $C_{0.5}$ | $C_{0.95}$ | 2-week ahead incident AE | WIS | $C_{0.5}$ | $C_{0.95}$ | 1-week ahead cumulative AE | WIS | $C_{0.5}$ | $C_{0.95}$ | 2-week ahead cumulative AE | WIS | $C_{0.5}$ | $C_{0.95}$ |
|---|---|---|---|---|---|---|---|---|---|---|---|---|---|---|---|---|
| epiforecasts-EpiExpert | 13,643 | 9,574 | 7/10 | 8/10 | 37,395 | 24,980 | 2/9 | 5/9 | 13,620 | 9,596 | 7/10 | 8/10 | 52,523 | 33,602 | 2/9 | 6/9 |
| epiforecasts-EpiNow2 | 11,006 | 7,041 | 5/10 | 7/10 | 38,906 | 25,308 | 2/9 | 6/9 | 11,028 | 7,048 | 5/10 | 7/10 | 47,373 | 30,303 | 2/9 | 7/9 |
| ICM-agentModel | | | 2/4 | 4/4 | | | 0/3 | 1/3 | | | | | | | 0/3 | 2/3 |
| ITWW-county_repro | 18,149 | 14,687 | 2/10 | 4/10 | 33,298 | 27,208 | 2/9 | 4/9 | 17,227 | 13,786 | 3/10 | 5/10 | 50,638 | 41,115 | 1/9 | 4/9 |
| LANL-GrowthRate | 15,956[a] | 9,490[a] | 3/7 | 7/7 | 49,295[a] | 27,220[a] | 1/6 | 6/6 | 15,269 | 9,311 | 3/10 | 10/10 | 62,801 | 36,562 | 3/9 | 8/9 |
| MIMUW-StochSEIR | 32,620[a] | 23,266[a] | 3/5 | 5/5 | 60,490[a] | 45,815[a] | 2/4 | 2/4 | | | | | | | | |
| MIT_CovidAnalytics-DELPHI | | | 2/9 | 5/9 | | | 1/8 | 4/8 | | | | | | | | |
| MOCOS-agent1 | 13,273 | 9,124 | 2/10 | 5/10 | 31,610 | 24,976 | 0/9 | 3/9 | 13,273 | 9,124 | 2/10 | 5/10 | 43,215 | 32,106 | 1/9 | 3/9 |
| SDSC_ISG-TrendModel | 7,633 | | | | | | | | 7,656 | | 2/5 | 5/5 | | | | |
| USC-SIkJalpha | 10,292 | | 0/1 | 1/1 | 24,138 | | 2/9 | | 13,560 | | 0/1 | 1/1 | 35,390 | | | |
| KIT-baseline | 28,164 | 18,119 | 5/10 | 9/10 | 52,890 | 35,107 | 2/9 | 6/9 | 28,235 | 18,154 | 5/10 | 9/10 | 80,765 | 53,656 | 2/9 | 6/9 |
| KIT-extrapolation_baseline | 18,311 | 11,917 | 6/10 | 10/10 | 55,060 | 34,212 | 3/9 | 7/9 | 18,289 | 11,912 | 6/10 | 10/10 | 76,607 | 45,935 | 3/9 | 8/9 |
| KIT-time_series_baseline | 22,497 | 14,100 | 5/10 | 10/10 | 60,079 | 37,980 | 5/9 | 9/9 | 22,475 | 14,098 | 5/10 | 10/10 | 84,530 | 52,303 | 4/9 | 9/9 |
| KITCOVIDhub-inverse_wis_ensemble | 12,768 | 8,456 | 4/10 | 10/10 | 36,229 | 24,628 | 3/9 | 6/9 | 11,865 | 7,733 | 4/10 | 10/10 | 44,477 | 30,643 | 2/9 | 6/9 |
| KITCOVIDhub-mean_ensemble | 12,982 | 8,320 | 3/10 | 9/10 | 36,338 | 23,598 | 3/9 | 6/9 | 12,051 | 7,582 | 5/10 | 9/10 | 44,254 | 29,461 | 2/9 | 7/9 |
| KITCOVIDhub-median_ensemble | 14,196 | 8,862 | 5/10 | 10/10 | 39,829 | 24,620 | 2/9 | 6/9 | 14,033 | 8,698 | 5/10 | 9/10 | 50,935 | 30,699 | 2/9 | 5/9 |

**Poland, deaths**

| Model | 1-week ahead incident AE | WIS | $C_{0.5}$ | $C_{0.95}$ | 2-week ahead incident AE | WIS | $C_{0.5}$ | $C_{0.95}$ | 1-week ahead cumulative AE | WIS | $C_{0.5}$ | $C_{0.95}$ | 2-week ahead cumulative AE | WIS | $C_{0.5}$ | $C_{0.95}$ |
|---|---|---|---|---|---|---|---|---|---|---|---|---|---|---|---|---|
| epiforecasts-EpiExpert | 285 | 176 | 4/10 | 10/10 | 605 | 374 | 1/9 | 8/9 | 285 | 176 | 4/10 | 10/10 | 874 | 530 | 3/9 | 8/9 |
| epiforecasts-EpiNow2 | 386 | 261 | 3/10 | 7/10 | 1,110 | 781 | 2/9 | 4/9 | 386 | 261 | 3/10 | 7/10 | 1,528 | 1,049 | 2/9 | 5/9 |
| ICM-agentModel | 752[a] | 672[a] | 2/8 | 3/8 | 1,881[a] | 1,237[a] | 0/7 | 2/7 | 1,178[a] | 727[a] | 1/8 | 4/8 | 2,955[a] | 1,865[a] | 0/7 | 3/7 |
| Imperial-ensemble2 | 397 | 238 | 3/10 | 7/10 | 701 | 613 | 0/9 | 2/9 | 369 | 211 | 3/10 | 8/10 | 1,219 | 1,085 | 0/9 | 2/9 |
| ITWW-county_repro | 525 | 484 | 0/10 | 1/10 | 834 | 529 | 2/9 | 7/9 | 524 | 483 | 0/10 | 1/10 | 637 | 404 | 3/9 | 8/9 |
| LANL-GrowthRate | 239[a] | 175[a] | 4/7 | 7/7 | 404[a] | 251[a] | 3/6 | 6/6 | 216 | 152 | 5/10 | 10/10 | | | | |
| MIMUW-StochSEIR | | | 1/5 | 4/5 | | | | | | | 1/5 | 4/5 | | | 0/4 | 4/4 |
| MIT_CovidAnalytics-DELPHI | 512[a] | 329[a] | 1/8 | 6/8 | 663[a] | 434[a] | 1/8 | 6/8 | 597[a] | 417[a] | 2/9 | 6/9 | 1,075[a] | 782[a] | 1/8 | 3/8 |
| MOCOS-agent1 | 194 | 147 | 9/10 | 10/10 | 420 | 272 | 7/9 | 8/9 | 194 | 147 | 9/10 | 10/10 | 556 | 382 | 7/9 | 9/9 |
| SDSC_ISG-TrendModel | 154 | | | | 240 | | | | 256 | | 0/1 | 1/1 | 242 | | | |
| USC-SIkJalpha | 206 | | 0/1 | 1/1 | | | | | | | | | | | | |
| KIT-baseline | 437 | 275 | 5/10 | 10/10 | 488 | 313 | 5/9 | 7/9 | 437 | 274 | 5/10 | 10/10 | 1,245 | 793 | 2/9 | 6/9 |
| KIT-extrapolation_baseline | 408 | 286 | 6/10 | 8/10 | 996 | 702 | 5/9 | 8/9 | 408 | 286 | 6/10 | 8/10 | 1,423 | 989 | 4/9 | 7/9 |
| KIT-time_series_baseline | 546 | 339 | 6/10 | 10/10 | 1,371 | 856 | 5/9 | 8/9 | 546 | 339 | 6/10 | 10/10 | 1,921 | 1,212 | 4/9 | 8/9 |
| KITCOVIDhub-inverse_wis_ensemble | 220 | 153 | 6/10 | 10/10 | 585 | 362 | 4/9 | 9/9 | 242 | 162 | 7/10 | 10/10 | 702 | 450 | 4/9 | 9/9 |
| KITCOVIDhub-mean_ensemble | 252 | 163 | 7/10 | 9/10 | | | | | 265 | 171 | 7/10 | 9/10 | 815 | 522 | 4/9 | 9/9 |
| KITCOVIDhub-median_ensemble | 215 | 148 | 6/10 | 10/10 | 471 | 289 | 2/9 | 9/9 | 231 | 160 | 6/10 | 10/10 | 707 | 458 | 4/9 | 9/9 |

[a]Entries where scores were imputed for at least 1 week. Weighted interval scores and absolute errors were imputed with the worst (largest) score achieved by any other forecast for the respective target and week. Models marked thus received a pessimistic assessment of their performance. If a model covered less than two-thirds of the evaluation period, results are omitted.

**Table 3 Forecast models contributed by independent external research teams.**

| Category | Model | Description | NPI | Test | Age | DE | PL | Truth | Pr |
|---|---|---|---|---|---|---|---|---|---|
| Agent-based | ICM-agentModel | Agent-based model for stochastic simulations of air-borne disease spread. Agents are assigned to geographically distributed contexts. The model implements a travel module that moves agents between cities[47]. | ✓ | ✓ | ✓ | | ✓ | JHU | ✓ |
| | MOCOS-agent1 | Agent-based model. Continuous-time stochastic microsimulation based on census data, including contact tracing, testing and quarantine[48]. Relevant duration time distributions are based on empirical data. | ✓ | ✓ | ✓ | | ✓ | JHU | ✓ |
| Compartment | CovidAnalytics-DELPHI[1] | Country-level modified SEIR model accounting for changing interventions and underdetection[49]. | ✓ | | | ✓ | ✓ | JHU | ✓ |
| | FIAS_FZJ-EpiIGer | Country-level deterministic model, extension of classical SEIR approach, takes explicitly into account undetected cases and reporting delays[50]. | | | | ✓ | | ECDC | ✓ |
| | LeipzigIMISE-SECIR | An extension of the SECIR type model implemented as input-output non-linear dynamical system. Joint fit of data on test positives, deaths, and ICU occupancy accounting for reporting delays. | ✓ | | | ✓ | | ECDC | ✓ |
| | MIMUW-StochSEIR | SEIR model with extensions: introduction of the undiagnosed compartment; testing limits influencing number of diagnosed cases; stochastic perturbations of time-dependent contact rate. | | | | | ✓ | JHU | ✓ |
| | UCLA-SuEIR[2] | A variant of the SEIR model considering both untested and unreported cases[51]. The model considers reopening and assumes the susceptible population will increase after the reopen. | ✓ | ✓ | | ✓ | | JHU | ✓ |
| | USC-SIkJalpha[3] | Reduces a heterogeneous rate model into multiple simple linear regression problems. True susceptible population is identified based on reported cases, whenever possible.[52] | | | | ✓ | ✓ | JHU | ✓ |
| Growth rate/ renewal eq. | epiforecasts-EpiNow2 | An exponential growth model that uses a time-varying $R_t$ trajectory to forecast latent infections, then convolves these using known delays to observations.[53]. Beyond the forecast horizon $R_t$ is assumed to be static. | | | | ✓ | ✓ | ECDC | ✓ |
| | SDSC_ISG-TrendModel[4] | Robust seasonal trend decomposition for smoothing of daily observations with further linear or multiplicative extrapolation. | | | | ✓ | ✓ | ECDC | ✓ |
| | ITWW-county_repro | Forecasts of county level incidence based on regional reproduction numbers estimated via small area estimation. | | | ✓ | ✓ | ✓ | ECDC | ✓ |
| | LANL-GrowthRate[5] | Dynamic SI model for cases with growth rate parameter updated at each model run (via regression model with day-of-week effect). The deaths forecast is a fraction of the cases forecasts (fraction learned via regression and updated at each run). | | | | ✓ | ✓ | JHU | ✓ |

**Table 3 (continued)**

| Category | Model | Description | NPI | Test | Age | DE | PL | Truth | Pr |
|---|---|---|---|---|---|---|---|---|---|
| Human judgement | epiforecasts- EpiExpert | A mean ensemble of predictions from experts and non-experts. Predictions are made via a web app[6] (ref. [54]) by choosing a type of distribution and specifying its median and width. | (✓) | (✓) | (✓) | ✓ | ✓ | ECDC | ✓ |
| Forecast ensemble | Imperial- ensemble2[7] | Unweighted average of three forecasts for death counts (see reference in footnote). | | | | ✓ | ✓ | ECDC | ✓ |

*NPI: Does the forecast model explicitly account for non-pharmaceutical interventions? Test: Does the model account for changing testing strategies? Age: Is the model age-structured? DE, PL: Are forecasts issued for Germany and Poland, respectively? Truth: Which truth data source does the model use? Pr: Are forecasts probabilistic (23 quantiles)?*
*Teams marked with footnotes run their own dashboards: ¹https://www.covidanalytics.io, ²https://covid19.uclaml.org, ³https://scc-usc.github.io/ReCOVER-COVID-19, ⁴https://renkulab.shinyapps.io/COVID-19-Epidemic-Forecasting, ⁵https://covid-19.bsvgateway.org,*
*⁶app.crowdforecastr.org, ⁷https://mrc-ide.github.io/covid19-short-term-forecasts*

setting the link between case counts and healthcare burden is expected to change, however, this decision will need to be re-assessed.

Estimation of total numbers of infected (including unreported) and effective reproductive numbers are other areas where a multi-model approach can be helpful (see ref. [36] for an example of the latter). While due to the lack of appropriate truth data these do not qualify as true prediction tasks, ensemble averages can again give a better picture of the associated uncertainty.

The German and Polish Forecast Hub will continue to compile short-term forecasts and process them into forecast ensembles. With the start of vaccine rollout and the emergence of new variants in early 2021, models face a new layer of complexity. We aim to provide further systematic evaluations for future phases, contributing to a growing body of evidence on the potential and limits of pandemic short-term forecasting.

## Methods
We now lay out the formal framework of our evaluation study. Unless stated differently, the described approach is the same as in the study protocol[24].

**Submission system and rhythm**. All submissions were collected in a standardised format in a public repository to which teams could submit (https://github.com/KITmetricslab/covid19-forecast-hub-de[23]). For teams running their own repositories, the Forecast Hub Team put in place software scripts to re-format forecasts and transfer them into the Hub repository. Participating teams were asked to update their forecasts on a weekly basis using data up to Monday. Submission was possible until Tuesday 3 p.m. Berlin/Warsaw time. Delayed submission of forecasts was possible until Wednesday, with exceptional further extensions possible in case of technical issues. Delays of submissions were documented (Supplementary Note 6).

**Forecast targets and format**. We focus on short-term forecasting of confirmed cases and deaths from COVID-19 in Germany and Poland 1 and 2 weeks ahead. Here, weeks refer to Morbidity and Mortality Weekly Report (MMWR) weeks, which start on Sunday and end on Saturday, meaning that 1-week-ahead forecasts were actually 5 days ahead, 2-week ahead forecasts were twelve days ahead, etc. All targets were defined by the date of reporting to the national authorities. This means that modellers have to take reporting delays into account, but has the advantage that data points are usually not revised over the following days and weeks. From a public health perspective there may be advantages in using data by symptom onset; however, for Germany, the symptom onset date is only available for a subset of all cases (50–70%), while for Poland no such data were publicly available during our study period. All targets were addressed both on cumulative and weekly incident scales. Forecasts could refer to both data from the European Centre for Disease Prevention and Control[25] (ECDC) and Johns Hopkins University Centre for Systems Science and Engineering[26] (JHU CSSE). In this article, we focus on the preregistered period of 12 October 2020 to 19 December 2020 (see Fig. 1). Note that on 14 December 2020, the ECDC data set on COVID-19 cases and deaths in daily resolution was discontinued. For the last weekly data point we therefore used data streams from Robert Koch Institute and the Polish Ministry of Health that we had previously used to obtain regional data and which up to this time had been in agreement with the ECDC data.

Most forecasters also produced and submitted 3- and 4-week-ahead forecasts (which were specified as targets in the study protocol). These horizons, also used in the US COVID-19 Forecast Hub[15], were originally defined for deaths. Owing to their lagged nature, these were considered predictable independently of future policy or behavioural changes up to 4 weeks ahead; see[37] for a similar argument. During the summer months, when incidence was low and intervention measures largely constant, the same horizons were introduced for cases. As the epidemic situation and intervention measures became more dynamic in autumn, it became clear that case forecasts further than 2 weeks (12 days) ahead were too dependent on yet unknown interventions and the consequent changes in transmission rates. It was therefore decided to restrict the default view in the online dashboard to 1- and 2-week-ahead forecasts only. At the same time we continued to collect 3- and 4-week-ahead outputs. Most models (with the exception of epiforecasts-EpiExpert, COVIDAnalytics-Delphi and in some exceptional cases MOCOS-agent1) do not anticipate policy changes, so that their outputs can be seen as "baseline projections", i.e., projections for a scenario with constant interventions. In accordance with the study protocol we also report on 3- and 4-week-ahead predictions, but these results have been deferred to Supplementary Note 8.

Teams were asked to report a total of 23 predictive quantiles (1%, 2.5%, 5%, 10%, …, 90%, 95%, 97.5%, 99%) in addition to their point forecasts. This motivates considering both forecasts of cumulative and incident quantities, as predictive quantiles for these generally cannot be translated from one scale to the other. Not

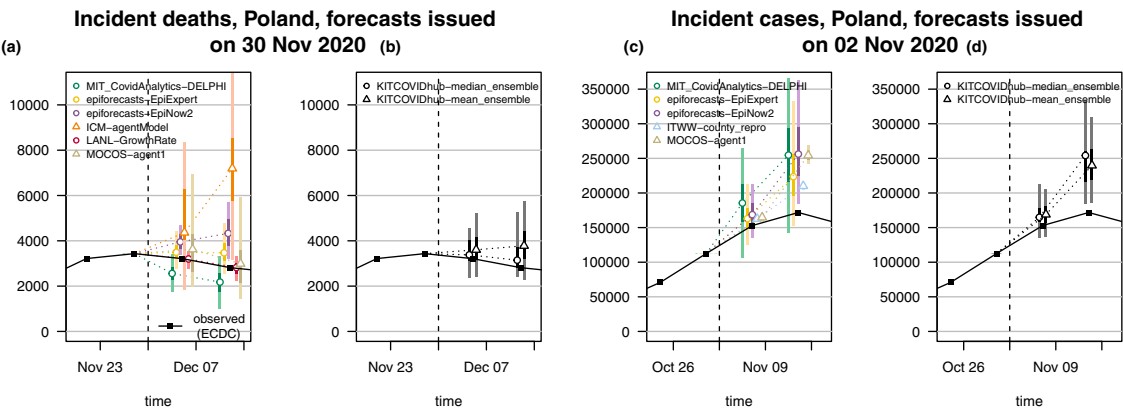

**Fig. 5 Examples of median and mean ensembles.** One- and 2-week-ahead forecasts of incident deaths in Poland issued on 30 November, and of incident cases in Poland issued on 2 November 2020. Panels (**a** and **c**) show the respective member forecasts, panels (**b** and **d**) the resulting ensembles. Both predictive medians and 95% (light) and 50% (dark) prediction intervals are shown. The dashed vertical line indicates the date at which the forecasts were issued.

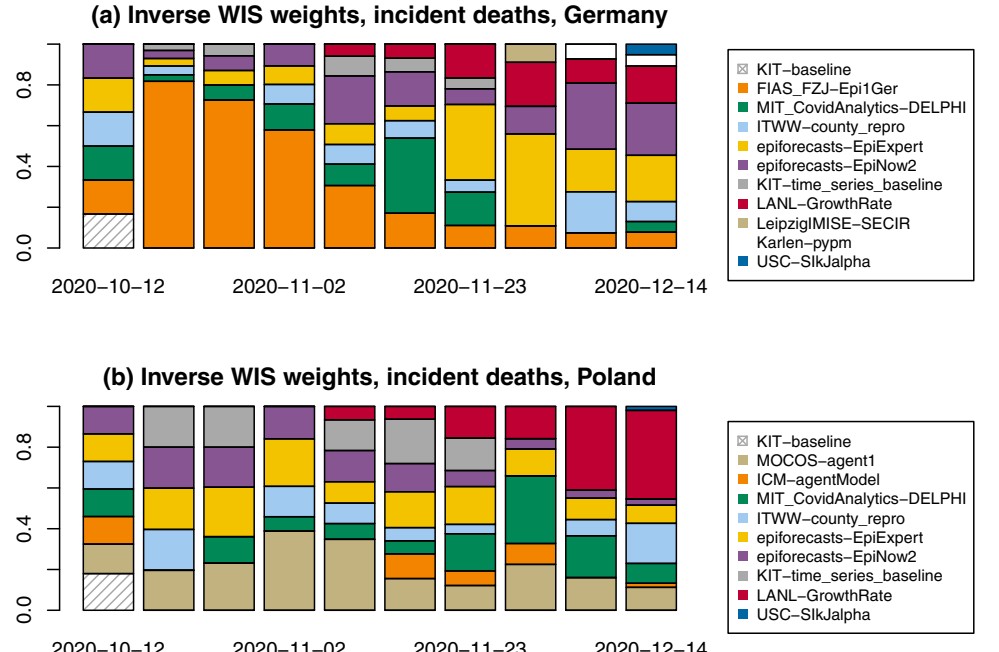

**Fig. 6 Examples of inverse WIS weights.** Inverse-WIS (weighted interval score) weights for forecasts of incident deaths in (**a**) Germany and (**b**) Poland.

all teams provided such probabilistic forecasts, though, and we also accepted pure point forecasts.

**Evaluation measures.** The submitted quantiles of a predictive distribution $F$ define 11 central prediction intervals with nominal coverage level $1 - \alpha$ where $\alpha = 0.02, 0.05, 0.10, 0.20, \ldots, 0.90$. Each of these can be evaluated using the interval score[38]:

$$\text{IS}_\alpha(F, y) = (u - l) + \frac{2}{\alpha} \times (l - y) \times \chi(y < l) + \frac{2}{\alpha} \times (y - u) \times \chi(y > u). \quad (1)$$

Here $u$ and $l$ are the lower and upper ends of the respective interval, $\chi$ is the indicator function and $y$ is the eventually observed value. The three summands can be interpreted as a measure of sharpness and penalties for under- and over-prediction, respectively. The primary evaluation measure used in this study is the weighted interval score[39] (WIS), which combines the absolute error (AE) of the predictive median $m$ and the interval scores achieved for the eleven nominal levels. The WIS is a well-known quantile-based approximation of the continuous ranked probability score[38] (CRPS) and, in the case of our 11 intervals, defined as

$$\text{WIS}(F, y) = \frac{1}{11.5} \times \left( \frac{1}{2} \times |y - m| + \sum_{k=1}^{11} \left( \frac{\alpha_k}{2} \times \text{IS}_{\alpha_k}(F, y) \right) \right), \quad (2)$$

where $\alpha_1 = 0.02, \alpha_2 = 0.05, \alpha_3 = 0.10, \alpha_4 = 0.20, \ldots, \alpha_{11} = 0.90$. Both the IS and

WIS are proper scoring rules[38], meaning that they encourage honest reporting of forecasts. The WIS is a generalisation of the absolute error to probabilistic forecasts. It reflects the distance between the predictive distribution $F$ and the eventually observed outcome $y$ on the natural scale of the data, with smaller values being better. As secondary measures of forecast performance we considered the absolute error (AE) of point forecasts and the empirical coverage of 50% and 95% prediction intervals. In this context we note that WIS and AE are equivalent for deterministic forecasts (i.e., forecasts concentrating all probability mass on a single value). This enables a principled comparison between probabilistic and deterministic forecasts, both of which appear in the present study. Applying the absolute error implies that forecasters should report predictive medians, as pointed out in the paper describing the employed evaluation framework[39].

In the evaluation we needed to account for the fact that forecasts can refer to either the ECDC or JHU data sets. We performed all forecast evaluations once using ECDC data and once using JHU data, with ECDC being our prespecified primary data source. For cumulative targets we shifted forecasts that refer to the other truth data source additively by the last observed difference. This is a pragmatic strategy to align forecasts with the last state of the respective time series.

A difficulty in comparative forecast evaluation lies in the handling of missing forecasts. For this case (which occurred for several teams) we prespecified that the missing score would be imputed with the worst (i.e., largest) score obtained by any other forecaster for the same target. The rationale for this was to avoid strategic omission of forecasts in weeks with low perceived predictability. In the respective

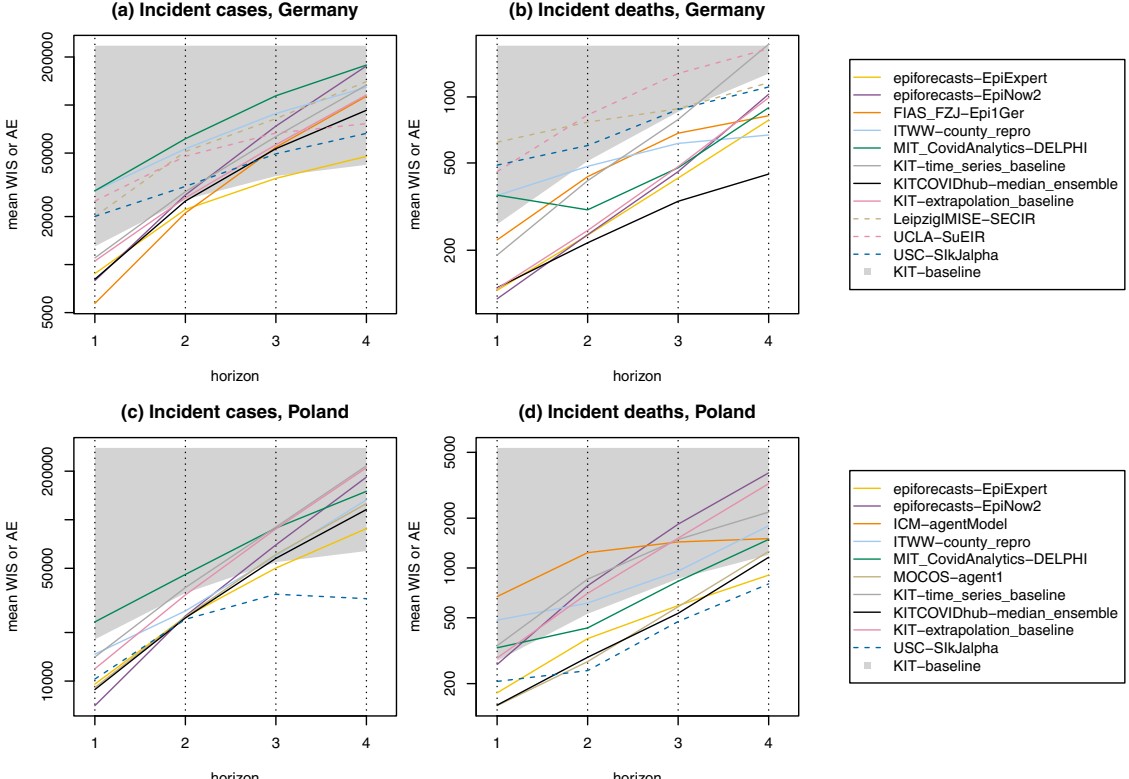

**Fig. 7 Forecast performance 1 through 4 weeks ahead.** Mean-weighted interval score (WIS) by target and prediction horizon in Germany (**a**, **b**) and Poland (**c**, **d**). We display submitted models and the preregistered median ensemble (logarithmic *y*-axis). For models providing only point forecasts, the mean absolute error (AE) is shown (dashed lines). The lower boundary of the grey area represents the baseline model KIT-baseline. Line segments within the grey area thus indicate that a model fails to outperform the baseline. The numbers underlying this figure can be found in Tables 1 and 2.

summary tables any such instances are marked. All values reported are mean scores over the evaluation period, though if more than a third of the forecasts were missing we refrain from reporting.

**Baseline forecasts**. In order to put evaluation results into perspective we use three simple reference models. Note that only the first was prespecified. The two others were added later as the need for comparisons to simple, but not completely naive, approaches was recognised. More detailed descriptions are provided in Supplementary Note 2.

*KIT-baseline*. A naive last-observation carried-forward approach (on the incidence scale) with identical variability for all forecast horizons (estimated from the last five observations). This is very similar to the null model used by Funk et al.[17].

*KIT-extrapolation baseline*. A multiplicative extrapolation based on the last two observations with uncertainty bands estimated from five preceding observations.

*KIT-time series baseline*. An exponential smoothing model with multiplicative error terms and no seasonality as implemented in the R package forecast[40] and used for COVID-19 forecasting by Petropoulos and Makridakis[41].

**Contributed forecasts**. During the evaluation period from October to December 2020, we assembled short-term predictions from a total of 14 forecast methods by 13 independent teams of researchers. Eight of these are run by teams collaborating directly with the Hub, based on models these researchers were either already running or set up specifically for the purpose of short-term forecasting. The remaining short-term forecasts were made available via dedicated online dashboards by their respective authors, often along with forecasts for other countries. With their permission, the Forecast Hub team assembled and integrated these forecasts. Table 3 provides an overview of all included models with brief descriptions and information on the handling of non-pharmaceutical interventions, testing strategies, age strata and the source used for truth data. More detailed verbal descriptions can be found in Supplementary Note 3. The models span a wide range of approaches, from computationally expensive agent-based simulations to human judgement forecasts. Not all models addressed all targets and forecast horizons suggested in our project; which targets were addressed by which models can be seen from Tables 1 and 2.

**Ensemble forecasts**. We assess the performance of three different forecast aggregation approaches:

*KITCOVIDhub-median ensemble*. The $\alpha$-quantile of the ensemble forecast for a given quantity is given by the median of the respective $\alpha$-quantiles of the member forecasts. The associated point forecast is the quantile at level $\alpha = 0.50$ of the ensemble forecast (same for other ensemble approaches).

*KITCOVIDhub-mean ensemble*. The $\alpha$-quantile of the ensemble forecast for a given quantity is given by the mean of the respective $\alpha$-quantiles of the member forecasts.

*KITCOVIDhub-inverse WIS ensemble*. The $\alpha$-quantile of the ensemble forecast is a weighted average of the $\alpha$-quantiles of the member forecasts. The weights are chosen inversely to the mean WIS value obtained by the member models over six recently evaluated forecasts (last three 1-week-ahead, last two 2-week-ahead, last 3-week-ahead; missing scores are again imputed by the worst score achieved by any model for the respective target). This is done separately for incident and cumulative forecasts. The inverse-WIS ensemble is a pragmatic strategy to base weights on past performance, which is feasible with a limited amount of historical forecast/observation pairs (see[42] for a similar approach).

Only models providing complete probabilistic forecasts with 23 quantiles for all four forecast horizons were included into the ensemble for a given target. It was not required that forecasts be submitted for both cumulative and incident targets, so that ensembles for incident and cumulative cases were not necessarily based on exactly the same set of models. The Forecast Hub Team reserved the right to screen and exclude member models in case of implausibilities. Decisions on inclusion were taken simultaneously for all three ensemble versions and were documented in the Forecast Hub platform (file decisions_and_revisions.txt in the main folder of the repository). The main reasons for the exclusion of forecasts from the ensemble were forecasts in an implausible order of magnitude or forecasts with vanishingly small or excessive uncertainty. As it showed comparable performance to submitted forecasts, the KIT-time_series_baseline model was included in the ensemble forecasts in most weeks.

Preliminary results from the US COVID-19 Forecast Hub indicate better forecast performance of the median compared to the mean ensemble[43], and the median ensemble has served as the operational ensemble since 28 July 2020. Up to date, trained ensembles yield only limited, if any, benefits[30]. We therefore

prespecified the median ensemble as our main ensemble approach. Note that in other works[19,44], ensembles have been constructed by combining probability densities rather than quantiles. These two approaches have somewhat different behaviour, but no general statement can be made which one yields better performance[45]. As in our setting member forecasts were reported in a quantile format we resort to quantile-based methods for aggregation.

**Reporting summary**. Further information on research design is available in the Nature Research Reporting Summary linked to this article.

## Data availability
The forecast data generated in this study have been deposited in a GitHub repository (https://github.com/KITmetricslab/covid19-forecast-hub-de), with a stable Zenodo release available under accession code 4752079 (https://doi.org/10.5281/zenodo.4752079,[23]). This repository also contains all truth data used for evaluation. Details on how truth data were obtained can be found in Supplementary Note 4. Forecasts can be visualised interactively at https://kitmetricslab.github.io/forecasthub/. Source data to reproduce Figs. 1–7 are provided with this paper.

## Code availability
Codes to reproduce figures and tables are available at https://github.com/KITmetricslab/analyses_de_pl, with a stable version at https://doi.org/10.5281/zenodo.5085398[46]. The results presented in this paper have been generated using the release "revision1" of the repository https://github.com/KITmetricslab/covid19-forecast-hub-de, see above for the link to the stable Zenodo release.

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

## Acknowledgements

We are grateful to the team of the US COVID-19 Forecast Hub, in particular Evan L. Ray and Nicholas G. Reich, for fruitful exchange and their support. We would like to thank Dean Karlen for contributions to the Forecast Hub from December 2020 onwards and Berit Lange for helpful comments. We moreover want to thank Fabian Eckelmann and Knut Perseke who implemented the interactive visualisation tool. The work of J. B. was supported by the Helmholtz Foundation via the SIMCARD Information and Data Science Pilot Project. S.B. acknowledges support from the Wellcome Trust (219415). N.I.B. acknowledges funding by the Health Protection Research Unit (grant code NIHR200908). S.F. and S.A. acknowledge support from the Wellcome Trust (grant no. 210758/Z/18/Z). The work of A. S. was supported by National Science Foundation Award No. 2027007 (RAPID). T.G. and D.W. are grateful for support by the Klaus Tschira Foundation. The content is solely the responsibility of the authors and does not necessarily represent the official views of the institutions they are affiliated with.

## Author contributions

J.B., T.G. and M. Schienle conceived the study with advice from A.U. J.B., D.W., J.D., K. Görgen and J.L.K. put in place and maintained the forecast submission and processing system. A.U. coordinated the creation of an interactive visualisation tool. J.B. performed the evaluation analyses with input from D.W., T.G., M. Schienle and members of various teams. S.A., M.V.B., D.B., S.B., M.B., N.I.B., J.P.B., L.C., G.F., J.F., S.F., K. Gogolewski, Q.G., S.H., T.H., Y.K., H.K., T.K., E.K., M.L.L., J.H.M., I.J.M., K.N., T.O., F.R., M. Scholz, S.S., A.S., J.Z. and D.Z. contributed forecasts (see list of contributors by team). J.B. and M. Schienle wrote the manuscript, with T.G. providing substantial editing, and T.K., M.B. and K. Gogolewski contributing to the description of intervention measures in Poland. All teams and members of the coordinating team provided descriptions of the respective models as well as feedback on the manuscript.

## Funding

## Competing interests

The authors declare no competing interests.

## Additional information

## List of Contributors by Team

**CovidAnalytics-DELPHI** Michael Lingzhi Li[17], Dimitris Bertsimas[6], Hamza Tazi Bouardi[6], Omar Skali Lami[6] & Saksham Soni[6]

**epiforecasts-EpiExpert and epiforecasts-EpiNow2** Sam Abbott[4], Nikos I. Bosse[4] & Sebastian Funk[4]

**FIAS FZJ-Epi1Ger** Maria Vittoria Barbarossa[5], Jan Fuhrmann[5,11] & Jan H. Meinke[11]

**German and Polish Forecast Hub Coordination Team** Johannes Bracher[1,2], Jannik Deuschel[1], Tilmann Gneiting[2,21], Konstantin Görgen[1], Jakob Ketterer[1], Melanie Schienle[1], Alexander Ullrich[3] & Daniel Wolffram[1,2]

**ICM-agentModel** Łukasz Górski[19], Magdalena Gruziel-Słomka[19], Artur Kaczorek[19], Antoni Moszyński[19], Karol Niedzielewski[19], Jedrzej Nowosielski[19], Maciej Radwan[19], Franciszek Rakowski[19], Marcin Semeniuk[19], Jakub Zieliński[19], Rafał Bartczuk[19,24] & Jan Kisielewski[19,25]

**Imperial-ensemble2** Sangeeta Bhatia[7]

**ITWW-county repro** Przemyslaw Biecek[26], Viktor Bezborodov[8], Marcin Bodych[8], Tyll Krueger[8], Jan Pablo Burgard[9], Stefan Heyder[14] & Thomas Hotz[14]

**LANL-GrowthRate** Dave A. Osthus[18], Isaac J. Michaud[18], Lauren Castro[10] & Geoffrey Fairchild[10]

**LeipzigIMISE-SECIR** Yuri Kheifetz[15], Holger Kirsten[15] & Markus Scholz[15]

**MIMUW-StochSEIR** Anna Gambin[12], Krzysztof Gogolewski[12], Błażej Miasojedow[12], Ewa Szczurek[12], Daniel Rabczenko[27] & Magdalena Rosińska[27]

**MOCOS-agent1** Marek Bawiec[8], Viktor Bezborodov[8], Marcin Bodych[8], Tyll Krueger[8], Tomasz Ożański[8], Barbara Pabjan[8], Ewaryst Rafajłowicz[8], Ewa Skubalska-Rafajłowicz[8], Wojciech Rafajłowicz[8], Przemyslaw Biecek[26], Agata Migalska[8,28] & Ewa Szczurek[29]

**SDSC ISG-TrendModel** Antoine Flahault[22], Elisa Manetti[22], Christine Choirat[16], Benjamin Bejar Haro[16], Ekaterina Krymova[16], Gavin Lee[16], Guillaume Obozinski[16], Tao Sun[16] & Dorina Thanou[23]

**UCLA-SuEIR** Quanquan Gu[13], Pan Xu[13], Jinghui Chen[13], Lingxiao Wang[13], Difan Zou[13] & Weitong Zhang[13]

**USC-SIkJalpha** Ajitesh Srivastava[30], Viktor K. Prasanna[30] & Frost Tianjian Xu[30]

[22]Institute of Global Health, Faculty of Medicine, University of Geneva, Geneva, Switzerland. [23]Center for Intelligent Systems, EPFL, Lausanne, Switzerland. [24]Institute of Psychology, John Paul II Catholic University of Lublin, Lublin, Poland. [25]Faculty of Physics, University of Białystok, Białystok, Poland. [26]Warsaw University of Technology, Warsaw, Poland. [27]Polish National Institute of Public Health—National Institute of Hygiene, Wroclaw, Poland. [28]Nokia Solutions and Networks, Wroclaw, Poland. [29]University of Warsaw, Warsaw, Poland. [30]University of Southern California, Los Angeles, USA.

