## [Peer Review File · Nature Communications]

REVIEWER COMMENTS

Reviewer #1 (Remarks to the Author):

This paper provides a description of a forecasting project done at the end of 2020 for COVID-19 cases and deaths in Germany and Poland. The authors describe the project design, the types of forecasts submitted, the creation of an ensemble of these forecasts, and their evaluation of the forecasts. They clearly outline the challenges of forecasting in the midst of the pandemic, as well as the utility of looking at several forecasts instead of relying solely on one. It is a well written paper that will serve as a useful resource for those planning to generate forecasts and for those interested in learning about best practices for interpreting or evaluating forecasts. I enjoyed reading the paper and only have a few suggested comments.

-Figure 1: Do the dashed lines correspond to the important changes in interventions and testing? If so, it would be helpful to add that to the legend (a little confusing since there aren't dashed lines for the times outside of the study period).

-Figure 1: Would recommend switching the monthly graphs to the top so the period of interest graphs are closer to the explanations of the measures.

-It would be helpful to explain the rationale for imputing missing forecasts as opposed to excluding them from the analysis.

-It might be helpful to have subsections for the different paragraphs in 4.1. I also think this section could be slightly streamlined.

-The numbers and date for bulk reporting in Poland in Figure 1 are slightly different than in section 4.1.

Reviewer #2 (Remarks to the Author):

This is an excellent paper providing an informative and thoughtful analysis of the COVID-19 forecasts produced for Germany and Poland. The companion github repository aids reproducibility which is to be applauded.

I have a few minor questions, which I hope lead to small improvements.

- One problem with modelling cases by reporting date is that the reporting processes will change over time. Did any models allow for this?
- Another problem with forecasting cases by reporting date is that it is less useful to public health authorities who should be more concerned with cases by date of onset. Perhaps this can be commented on?
- Were participants told that point forecasts should correspond to medians of the forecast distributions (as implied by the AE evaluation)?
- At least one country (Australia) has created an ensemble forecast of COVID-19 cases by combining the forecast distributions from individual models to create a mixture distribution. This is closer to the Reich et al approach to influenza forecasting, rather than taking medians of quantiles as done here. See https://www.doherty.edu.au/uploads/content_doc/Technical_report_4_July17.pdf for a tech report describing this approach.
- How were the forecasts used by public health authorities? In particular, were specific quantiles more important in determining policy responses?

Rob Hyndman

Reviewer #3 (Remarks to the Author):

The paper applies different statistical measures (scores) on different forecasting time series covering a period of about 2 months. The forecasting data was generated from different research groups using different models and collected in realtime in a well-designed, prespecified, and documented process. Indeed this is "...a substantial strength of our study...". Especially important is, that criteria for evaluation were communicated in advance. The paper gives a very good and forward-looking first idea for future processes of how to deal with the rising number of models and the question, how they can be integrated into decision sciences.

The idea of using distinct reliable measures for quantifying the performance of forecasting approaches is an important aspect of the current research on the COVID-19 pandemic. A strong focus is put on estimates of the predictive quality of the models. Furthermore, possible improvement of accuracy in ensemble forecasts is discussed.

The forecasting models themselves and their technical background is not the key concern of the paper. A good overview is given in Table 1 (see suggestions below). The investigation in the paper could benefit from a more detailed discussion of individual approaches on a conceptual and technical level, depending on the goal of the paper.

Another possibility is to describe in more detail, why "Forecasting is one of the key purposes of epidemic modeling, and despite being related to the understanding of underlying mechanisms, it is a conceptually distinct task". It could be described in more detail, why it is distinct from explanatory modeling or understanding of the natural system or it should be clearly described, why detailed method descriptions are not necessary in this paper. Most of the comments below are focusing on these questions.

In Table 1 forecasting methods are labeled e.g. with "Is the model age-structured?". This is restricted to a boolean value and gives a good overview of the models. But such features should probably be discussed more extensively. In turn, this leads to the question of which modeling or forecasting approaches are particularly suitable to predict/reproduce certain phenomena in the truth data.

In the discussion, the authors state that "different models may in fact be particularly suitable for different phases of an epidemic". Is it possible to correlate specific technical features or model complexity (that is partially provided by the included models) with predictive accuracy in different phases of the epidemic? The online dashboard referenced in the manuscript provides partial forecasting data for a longer period of time.

On page 14 it is mentioned "Certain models were originally conceived for what-if projections and retrospective assessments of longer-term dynamics and interventions. This focus on a global fit may limit their flexibility to align closely with the most recent data, making them less successful at short forecast horizons compared to simpler extrapolation approaches."

To what extent and in which aspects of spreading dynamics does increase model complexity/fidelity adds to forecasting quality? Or do the results of this meta-study suggest that model complexity is irrelevant or obstructive for short-term predictive power? Hence, indicating that purely statistical or agnostic approaches are sufficient or even better? Those are interesting and pressing questions. It is clear that these questions cannot be explained in detail in the paper, but they should be addressed in

a little bit more detail (and earlier in the paper).

There exists a large literature on epidemic models that was published before the current COVID pandemic and without relying on the currently available data. Often very clear conclusions about the relevance of certain modeling aspects have been made. How are such results reflected in the presented survey and in the forecasting models investigated?

One could expect that besides the more detailed description of methods (see considerations above) also particular aspects of the modeling process, like calibration or implemented algorithms are included in the survey. The corresponding section 3.2 "Contributed forecasts" should be extended a little bit.

The measures applied to seem plausible and were already tested in previous studies. A discussion of the measures is provided in section "Evaluation measures". However, this discussion could be a little bit more extensive.

The relatively short period of observation, which only covers the onset of an epidemic wave is understandable, should be briefly addressed (as it is described that work will continue)

Further, discuss the possible application in the decision-making process. How can uncertainty and misinterpretation be avoided when diverging predictions are made by different forecasting models. What is necessary (besides "finer geographical resolution")? What is the general conclusion about COVID-19 forecasting? E.g. Which figures (besides cases and deaths) should be investigated in the future? If possible briefly address the differentiation between reported and unreported cases.

The article provides detailed insight into the quality of a distinct set of forecasting algorithms. Maybe, to increase the significance of the results, the discussion can be extended to forecasting in general or - in a more granular fashion - to certain types of forecasting methods.

We would like to thank the review team for the constructive reports and helpful comments on our paper. In what follows we respond to the different points raised by the reviewers. The page references in the referees' statements refer to the original version. We also provide a version of the revised paper with track changes.

Please note that the paper has been re-structured in order to adhere to journal style (Introduction, Results, Discussion, Methods).

Response to Referee 1

This paper provides a description of a forecasting project done at the end of 2020 for COVID-19 cases and deaths in Germany and Poland. The authors describe the project design, the types of forecasts submitted, the creation of an ensemble of these forecasts, and their evaluation of the forecasts. They clearly outline the challenges of forecasting in the midst of the pandemic, as well as the utility of looking at several forecasts instead of relying solely on one. It is a well written paper that will serve as a useful resource for those planning to generate forecasts and for those interested in learning about best practices for interpreting or evaluating forecasts. I enjoyed reading the paper and only have a few suggested comments.

1: *Figure 1: Do the dashed lines correspond to the important changes in interventions and testing? If so, it would be helpful to add that to the legend (a little confusing since there aren't dashed lines for the times outside of the study period).*

Reply: Yes, the dashed lines correspond to changes in interventions and testing. We now clarify this in the caption of the figure and have added lines for events that precede the evaluation period.

2: *Figure 1: Would recommend switching the monthly graphs to the top so the period of interest graphs are closer to the explanations of the measures.*

Reply: We have moved the smaller graphs up as suggested.

3: *It would be helpful to explain the rationale for imputing missing forecasts as opposed to excluding them from the analysis.*

Reply: The motivation for imputing missing scores with the respective worst score was to avoid incentives to skip forecasts. If scores were just averaged over the available set of forecasts, forecasters might have gained an advantage by only covering weeks for which they expected to achieve good scores. In particular, as the weighted interval score typically grows with the order of magnitude of the forecasted quantities, there would have been an incentive to focus on weeks with lower incidence. We have added a brief remark on this to the section on evaluation measures.

4: *It might be helpful to have subsections for the different paragraphs in 4.1. I also think this section could be slightly streamlined.*

Reply: We have structured this section (now the beginning of the *Results* part of the paper) with paragraph headings, thereby following journal style, and we have shortened slightly.

5: *The numbers and date for bulk reporting in Poland in Figure 1 are slightly different than in section 4.1.*

Reply: Thank you for pointing out this error. We have corrected the erroneous date in Figure 1 (22,000 on the 24th of November is correct).

Response to Referee 2

This is an excellent paper providing an informative and thoughtful analysis of the COVID-19 forecasts produced for Germany and Poland. The companion github repository aids reproducibility which is to be applauded.

I have a few minor questions, which I hope lead to small improvements.

1,2: *One problem with modelling cases by reporting date is that the reporting processes will change over time. Did any models allow for this?*

Another problem with forecasting cases by reporting date is that it is less useful to public health authorities who should be more concerned with cases by date of onset. Perhaps this can be commented on?

Reply: We agree that aggregating counts by date of symptom onset has advantages from a public health perspective. The main reason for choosing data by date of reporting was the lack of complete data on the date of symptom onset:

- For Germany, the available counts by symptom onset date are actually a mix of counts by symptom onset, and counts by reporting date to a local authority, which is imputed whenever no disease onset date was reported. During our evaluation period, actual symptom onset dates were only available for 51% of the cases. So moving to this time scale may make things even more complex.

- For Poland only data by reporting date were publicly available during our evaluation period.

We have added a sentence that explains the limited availability of data by symptom onset to the section on *Forecast targets and format*.

In addition to the constraints imposed by data availability we see the following reasons for using data by reporting date:

- Data by reporting date (with few exceptions which did not occur during our study) are definitive immediately, and not subject to retrospective revision or completion. This makes them more suitable for timely forecast evaluation than data by symptom onset which typically take more than 10 days to stabilize (in Germany, the 95% quantile of the reporting delay was 13 days during our evaluation period).
- Counts by reporting date are commonly issued by German public media and evoked by politicians. We thus consider them a relevant measure that is typically taken into account by decision makers.

Only one model (ITWW-countyRepro) took into account data by symptom onset date, generated forecasts at this scale and then convoluted them to counts by reporting date via a delay distribution. This distribution was updated every week. Another team (Leipzig_IMISE) used to follow a similar approach, previous to the evaluation period, but switched to fitting directly to the time series by reporting date. All other models operated directly on the time series by reporting date. The more statistical models which quantify and extrapolate recent growth will only implicitly adapt to changing patterns in the observed time series. Most compartmental and microsimulation models involve an explicit reporting process (where applicable, the newly added extended model description in what is now Supplementary Section C contain details on the assumptions), but none of these attempted to assimilate data on changing delay distributions in real time.

For Germany we checked empirically how much reporting delays varied over our evaluation period. Figure 1 in this response shows the mean delay between symptom onset (where available) and reporting date to Robert Koch Institute. There is only moderate temporal variation around a mean delay of about seven days. We therefore believe that, while taking this aspect into account may lead to slight improvements of forecasts, the benefits are bound to be limited. For Poland no public data are available that would allow to assess reporting delays.

3: *Were participants told that point forecasts should correspond to medians of the forecast distributions (as implied by the AE evaluation)?*

Reply: Our preregistration document (<https://osf.io/zkdvb/>) states explicitly that absolute error will be used for the evaluation of point forecasts. While the protocol does not detail that this implies medians should be submitted, teams were repeatedly pointed to this aspect during discussions of evaluation results. Also, we pointed participants to (the then preprint version of) our paper on the employed evaluation framework (Bracher,

Figure 1: Average delay (in days) between date of symptom onset (where available) and reporting date in Germany.

Ray, Gneiting and Reich 2021; joint work with the US COVID-19 Forecast Hub). In Section 5 of this work the following is stated:

“As in the FluSight Challenge [7], the absolute error (AE) will be applied. This implies that teams should report the predictive median as a point forecast [32].”

We have added a remark on this aspect to the section on *Evaluation measures*.

4: *At least one country (Australia) has created an ensemble forecast of COVID-19 cases by combining the forecast distributions from individual models to create a mixture distribution. This is closer to the Reich et al approach to influenza forecasting, rather than taking medians of quantiles as done here. See https://www.doherty.edu.au/uploads/content_doc/Technical_report_4_July17.pdf for a tech report describing this approach.*

Reply: Thank you for pointing us to this report of which we were not aware. We have added a reference to the introduction and the final paragraph in the section on *Ensemble forecasts*.

5: *How were the forecasts used by public health authorities? In particular, were specific quantiles more important in determining policy responses?*

Reply: Our project is not formally funded or endorsed by German or Polish authorities, which is why we did not address this aspect explicitly in the manuscript. However, the three Polish groups who participated in the Forecast Hub (ICM, MOCOS and MIMUW) have provided consulting to the Polish Ministry of Health continuously since the start of the pandemic. We have been informed that the Ministry considers the short-term forecasts

by these groups as well as the ensemble forecast from the Forecast Hub. This, of course, is only one of many elements of the scientific input they receive, and the Polish groups also deliver much more detailed reports that are tailored to the requirements of the specific situation. While we do not have specific information on how the uncertainty measures are used by the ministry, there is a general awareness for the need to consult multiple models.

In Germany we have been collaborating with a team from the Robert Koch Institute (RKI, the national public health institute) that specializes in the analysis and interpretation of surveillance data. Their main interest is in overall trends (are the numbers going up or down, trends likely to change?), and typically 50% and 95% prediction intervals are considered. These are also in use for nowcasts produced by RKI (see e.g. https://www.rki.de/DE/Content/Infekt/EpidBull/Archiv/2020/Ausgaben/17_20.pdf).

Response to Referee 3

The paper applies different statistical measures (scores) on different forecasting time series covering a period of about 2 months. The forecasting data was generated from different research groups using different models and collected in realtime in a well-designed, prespecified, and documented process. Indeed this is "... a substantial strength of our study". Especially important is, that criteria for evaluation were communicated in advance. The paper gives a very good and forward-looking first idea for future processes of how to deal with the rising number of models and the question, how they can be integrated into decision sciences.

The idea of using distinct reliable measures for quantifying the performance of forecasting approaches is an important aspect of the current research on the COVID-19 pandemic. A strong focus is put on estimates of the predictive quality of the models. Furthermore, possible improvement of accuracy in ensemble forecasts is discussed.

1: *The forecasting models themselves and their technical background is not the key concern of the paper. A good overview is given in Table 1 (see suggestions below). The investigation in the paper could benefit from a more detailed discussion of individual approaches on a conceptual and technical level, depending on the goal of the paper.*

Reply: We agree that detailed information on the individual models will provide a more complete picture of our project. We have added extended descriptions of the forecasting models in the new Supplementary Section C.

2: *Another possibility is to describe in more detail, why "Forecasting is one of the key purposes of epidemic modeling, and despite being related to the understanding of underlying mechanisms, it is a conceptually distinct task". It could be described in more detail, why it is distinct from explanatory modeling or understanding of the natural system or it should be clearly described, why detailed method descriptions are not necessary in this paper. Most of the comments below are focusing on these questions.*

Reply: Explanatory models aim to describe the hidden mechanisms behind observable patterns in a transparent and biologically meaningful fashion. Predictive modelling, on

the other hand, is focused exclusively on accuracy with respect to observable quantities, such as cases and deaths. As argued by Keeling and Rohani (2008, pp. 8–9), transparency and accuracy can be conflicting goals that might favour distinct modelling strategies. Short-term accuracy can often be achieved by flexible, purely data-driven phenomenological approaches. These do not have any mechanistic underpinning and thus do not provide any insights on the underlying process of disease spread. Explanatory models, on the other hand, tend to be tailored to specific aspects of a system, and do not generally aim to quantitatively reproduce its behaviour as a whole.

The question of purely predictive and explanatory modelling is closely linked to the distinction between forecasting and scenario modelling. Scenario modelling aims to distinguish the differences between distinct hypothetical settings and thus necessarily makes statements about causes and effects. This requires interpretable explanatory models. Pure forecasts may well benefit from a detailed understanding of the mechanics of an epidemic, but this is not the primary goal.

We have extended the discussion of these facets within the *Introduction* section, have clarified the link to forecasting and scenario modelling, and have added a reference to the work by Baker et al. (2018), where these questions are discussed at length in the context of biological modelling.

3: *In Table 1 forecasting methods are labeled e.g. with “Is the model age-structured?”. This is restricted to a boolean value and gives a good overview of the models. But such features should probably be discussed more extensively.*

Reply: The extended model descriptions in the new Supplementary Section C provide more detailed descriptions of these aspects. Specifically, the `ITWW-county_repro` model uses one-year strata for Poland and the six age groups used by Robert Koch Institute in its data releases for Germany (0–4, 5–14, 15–34, 35–59, 60–79, 80+). The agent-based `ICM-agentModel` and `MOCOS-agent1` approaches assign each individual in their simulation an age based on census data. They thus closely mimic the age structure of the Polish population.

4: *In turn, this leads to the question of which modeling or forecasting approaches are particularly suitable to predict/reproduce certain phenomena in the truth data. In the discussion, the authors state that “different models may in fact be particularly suitable for different phases of an epidemic”. Is it possible to correlate specific technical features or model complexity (that is partially provided by the included models) with predictive accuracy in different phases of the epidemic? The online dashboard referenced in the manuscript provides partial forecasting data for a longer period of time.*

Reply: In Supplementary Section G, which is referenced in what is now the section on *Formal forecast evaluation* in the main paper, we have added Supplementary Figures 11–14 with the weekly WIS or absolute error achieved by the different models (at a one week lead time). Despite the overall picture being somewhat unwieldy, given the strong fluctuations in relative performance, interesting findings can be discerned:

- The ensemble model `KITCOVIDhub-median_ensemble` shows lower variability in relative performance than most contributed models, in particular for death forecasts.

Figure 2: Number of submitted models (excluding baseline and ensemble forecasts) for Germany (left) and Poland (right) by forecast date.

It constitutes the most reliable approach in terms of outperforming the baseline model.

- The human judgement approach **epiforecasts-EpiExpert** tends to perform favourably subsequently to change points in trends, as it avoids strong overshooting.
- The renewal equation models **epiforecasts-EpiNow2**, **ITWW-county_repro** and **LANL-GrowthRate**, on the other hand, show a rather strong tendency to overshoot at inflection points (see, e.g., cases in Germany, 6 November, and deaths in Poland, 13 November). This appears to make substantial sense, given the direct extrapolation approach.

While further patterns may seem discernable we believe that these would be speculative, given the limited length of the evaluation period and the pronounced fluctuation.

Figure 2 in this response shows how the number of submitted models (excluding baselines and ensembles) evolved over time. As the referee points out, for some models the records go back further than October, but for most teams (in particular, for **FIAS_FZJ-Epi1Ger**, **epiforecasts-EpiNow2**, **epiforecasts-EpiExpert**, **ITWW-county_repro**, **MIT_CovidAnalytics-DELPHI**, **ICM-agentModel** and **MIMUW-StochSEIR**) the beginning of the evaluation period was the starting point for weekly and complete submissions. In this light, we decided not to include data from before the evaluation period in these exploratory analyses.

We have not included data from the time after the current study period either because we consider it a meaningful – and preregistered – temporal delimitation. In particular, the study period is followed by three weeks of unreliable data over the Christmas and New Year holiday period, and subsequently a changed epidemiological situation characterized by the emergence of new variants and the start of vaccination campaigns. We therefore find it preferable to assess these periods separately and with appropriate contextualization, which is outside the scope of the present paper.

5: On page 14 it is mentioned “Certain models were originally conceived for what-if projections and retrospective assessments of longer-term dynamics and interventions. This

focus on a global fit may limit their flexibility to align closely with the most recent data, making them less successful at short forecast horizons compared to simpler extrapolation approaches.”

To what extent and in which aspects of spreading dynamics does increase model complexity/fidelity adds to forecasting quality? Or do the results of this meta-study suggest that model complexity is irrelevant or obstructive for short-term predictive power? Hence, indicating that purely statistical or agnostic approaches are sufficient or even better? Those are interesting and pressing questions. It is clear that these questions cannot be explained in detail in the paper, but they should be addressed in a little bit more detail (and earlier in the paper).

There exists a large literature on epidemic models that was published before the current COVID pandemic and without relying on the currently available data. Often very clear conclusions about the relevance of certain modeling aspects have been made. How are such results reflected in the presented survey and in the forecasting models investigated?

Reply: We have added a paragraph on these interesting and highly relevant issues to the *Discussion* section, where we refer to recent work by Johansson et al. (2019), McGowan et al. (2019) and Reich et al. (2019a). These papers find slightly better performance for statistical than for mechanistic models. However, they all cover seasonal diseases, and there is reason to believe that mechanistic models have benefits for emerging diseases, where historical data is limited and changing interventions shape the course of an epidemic outbreak.

In exploratory analyses of the performance of the forecasts in our system (Supplementary Figure 15, referenced in the section on *Formal forecast evaluation*) we have not been able to identify any clear patterns indicating that one general modelling approach might be superior to others. Here we categorized models following the scheme from (what is now) Table 3, which at least for agent-based, compartmental and renewal equation models implies an ordering by complexity. We believe that a large number of factors, including the applied inference approach, tuning based on expert knowledge, and manual plausibility checking impact model performance, but the accumulated data basis does not allow us to discern these. We are currently collaborating with colleagues from the US COVID-19 Forecast Hub to develop statistical methodologies to assess the impact of certain modelling choices on forecast performance as well as the similarity of forecasts produced by different models. However, we consider this to be outside the scope of the current paper.

6: One could expect that besides the more detailed description of methods (see considerations above) also particular aspects of the modeling process, like calibration or implemented algorithms are included in the survey. The corresponding section 3.2 “Contributed forecasts” should be extended a little bit.

Reply: The newly added detailed model descriptions in Supplementary Section C include statements on the statistical methodology/algorithms used for model calibration. These are quite diverse and include least-squares, maximum likelihood and Bayesian approaches. Methods employed to generate uncertainty intervals include re-sampling of

past errors and various forward sampling schemes.

7: *The measures applied to seem plausible and were already tested in previous studies. A discussion of the measures is provided in section “Evaluation measures”. However, this discussion could be a little bit more extensive.*

Reply: We have added the following points to provide a better understanding of the employed scores:

- We now mention that the WIS is a so-called proper scoring rule (Gneiting and Raftery 2007). Proper scoring rules are constructed such that forecasters optimize their expected score by reporting their true beliefs about the future, so that honest forecasting is incentivized.
- We also mention that the WIS is a direct generalization of the absolute error and thus interpretable on the natural scale of the data.
- Finally, we note the fact that using the absolute error means that predictive medians should be reported.

8: *The relatively short period of observation, which only covers the onset of an epidemic wave is understandable, should be briefly addressed (as it is described that work will continue).*

Reply: We agree that the period of observation is relatively short, but consider it a meaningful temporal subset of the pandemic. It coincides almost perfectly with the second wave in both countries, which was still unaffected by vaccination and caused by the original “wild-type” variant of the virus. This is followed by three weeks of rather unreliable data from the Christmas and New Year holiday period, and subsequently a new phase of the pandemic increasingly characterized by the shift towards the novel B.1.1.7 variant. The share of B.1.1.7 has been estimated at 6% in Germany and 9% in Poland already in the first half of February (European Centre for Disease Prevention and Control 2021) and has rapidly increased since. In parallel, in both countries vaccination started in the last week of December. We have added an explanation to the *Introduction* section.

9: *Further, discuss the possible application in the decision-making process. How can uncertainty and misinterpretation be avoided when diverging predictions are made by different forecasting models. What is necessary (besides “finer geographical resolution”)? What is the general conclusion about COVID-19 forecasting? E.g. Which figures (besides cases and deaths) should be investigated in the future? If possible briefly address the differentiation between reported and unreported cases.*

Reply: We believe that systematic documentation and evaluation of forecasts is an important step in order to make the outputs of models useful for decision making. There is value in reducing uncertainty where possible, and appropriately characterize it where it remains. Multi-model approaches make a valuable contribution here as single models often underestimate their own uncertainty (as documented by the below-nominal interval

coverage rates in Figure 3). The agreement – or lack thereof – of different models can then give a more realistic idea of how uncertain the future is.

As argued in the Discussion section, ensemble forecasts are an established tool in order to capitalize from multiple models, aggregate predictive information, and facilitate communication to decision makers. They have repeatedly been found to yield more consistent forecast performance over time (e.g. Reich 2019), a finding also present in our study (see Supplementary Figures 11–14 and discussion under point 4 of the response letter).

Several additional forecast targets could be investigated. In Germany we have had an exchange with a collaboratory of university hospitals aiming to generate forecasts of hospitalization numbers and ICU need in their institutions (Polotzek et al. 2020), but closer collaboration has been hampered by data privacy issues. We are now involved in a European initiative led by the European Centre for Disease Prevention and Control (ECDC), which currently covers cases and deaths, but is likely to include hospitalizations in the near future. Multi-model approaches can also be beneficial in the estimation of reproductive numbers and total numbers of infections (including unreported). Both differ from our current targets in that they are not actually observable, meaning that results cannot be evaluated against a later observed truth value. Nonetheless, taking into account several independently made estimates can help characterize the respective uncertainties. Indeed, some of our co-authors have led a multi-team R_t estimation project in the United Kingdom (see the added reference to UK Department of Health and Social Care, 2021), and the inclusion of this quantity in the European Forecast Hub is currently under discussion. Concerning total prevalence estimation, e.g. the UCLA team provides results in its online dashboard (<https://covid19.uclaml.org/>). In the paper, we have supplemented the respective material in the penultimate paragraph of the *Discussion* section.

10: The article provides detailed insight into the quality of a distinct set of forecasting algorithms. Maybe, to increase the significance of the results, the discussion can be extended to forecasting in general or – in a more granular fashion – to certain types of forecasting methods.

Reply: In order to add a more general perspective on the challenges of disease forecasting we have added a brief paragraph on important differences to weather forecasting. These include generally lower data quality and resolution, but also a poorer understanding of the underlying principles, in particular with respect to the social dynamics shaping an epidemic. We refer to the works by Moran et al. (2016) and Funk et al. (2009) for a more detailed discussion of these issues.

Further changes

Below we list further changes to the manuscript which we have made independently of the reviewers' comments.

1. The introduction needed to be re-structured in order to comply with the journal style.
2. In consultation with related forecasting projects (the US COVID-19 Forecast Hub and a non-public project in the UK) it was decided to slightly revise the definition of the weighted interval score, namely, from

$$\text{WIS}(F, y) = \frac{1}{12} \times \left(|y - m| + \sum_{k=1}^{11} \left(\frac{\alpha_k}{2} \times \text{IS}_{\alpha_k}(F, y) \right) \right)$$

to

$$\text{WIS}(F, y) = \frac{1}{11.5} \times \left(\frac{1}{2} \times |y - m| + \sum_{k=1}^{11} \left(\frac{\alpha_k}{2} \times \text{IS}_{\alpha_k}(F, y) \right) \right).$$

While both versions of the weighting can be motivated by an approximation to the CRPS, the new version is a better approximation for a finite number of intervals included. This new version has also been adopted in the recently published version of Bracher et al. (2021, <https://doi.org/10.1371/journal.pcbi.1008618>), as cited in the paper. The resulting changes in the reported values of the mean WIS are small and do not affect any of the interpretations.

3. Upon request from the respective team, the model previously called **Geneva-DeterministicGrowth** is now referred to as **SDSC-ISG_TrendModel**.
4. In the captions of Table 2 and (what used to be) Table 3 it was erroneously stated that $C_{0.95}$ denotes the coverage of 90% intervals, rather than 95% intervals. This has been corrected.
5. We have corrected one instance where a forecast for cases in Poland by the team **USC-SIkJalpha** had been processed erroneously from the group's repository. This correction has led to an improvement of the scores achieved by the respective model. The scores for the ensemble approaches are unaffected, as the **USC-SIkJalpha** team provided only point forecasts, which were not included in the ensembles.

Additional references

- European Centre for Disease Prevention and Control (2021): SARS-CoV-2: Increased circulation of variants of concern and vaccine rollout in the EU/EEA, 14th update. Available online at <https://www.ecdc.europa.eu/sites/default/files/documents/RRA-covid-19-14th-update-15-feb-2021.pdf>

- Polotzek, K., Karch, A., Karschau, J., von Wagner, M., Lünsmann, B., Menk, M., Römmele, C., and Schmitt, J. (2020): COVID-19-Pandemie: Regionale Steuerung der Patienten. Deutsches Ärzteblatt 118(3): A-84/B-74.

REVIEWERS' COMMENTS

Reviewer #3 (Remarks to the Author):

Thank you very much for including most of the suggested changes or discussing them.

I appreciate the improved introduction. The extended parts within the Introduction section clarify in a very good way, e.g. the differences between forecasting and scenario modeling. I do not agree with all the points, but it is well described and that's the way it should be...

Results and discussion sections are improved, with more detailed information and now do describe the idea of the publication in a more adequate way.

The new Supplementary Section C provides detailed descriptions for the model approaches and gives a good overview.

In Supplementary Section G additional information is provided, thank you very much for these descriptions.

I think it is also a good idea to match scores between the different forecast hubs as much as possible, as you did with the weighted interval score according to the US COVID-19 Forecast Hub. The usability of these projects improves a lot, if there are consistent with each other.

As the last suggestion, I would provide the thoughts about which outputs of models (and how) can be useful for decision making. You address the need to discuss ICU usage in the response to the referees' comments. The importance of such outcomes increases at the moment. If you could integrate these thoughts in the publication (even if I understand, that it is not the main goal), this would increase the benefit.

We would like to thank the review team for the constructive reports and helpful comments on our paper. In what follows we respond to the different points raised by the reviewers. The page references in the referees' statements refer to the original version. We also provide a version of the revised paper with track changes.

Response to Referee 3

Thank you very much for including most of the suggested changes or discussing them.

1: *I appreciate the improved introduction. The extended parts within the Introduction section clarify in a very good way, e.g. the differences between forecasting and scenario modeling. I do not agree with all the points, but it is well described and that's the way it should be.*

2: *Results and discussion sections are improved, with more detailed information and now do describe the idea of the publication in a more adequate way.*

3: *The new Supplementary Section C provides detailed descriptions for the model approaches and gives a good overview.*

4: *In Supplementary Section G additional information is provided, thank you very much for these descriptions.*

5: *I think it is also a good idea to match scores between the different forecast hubs as much as possible, as you did with the weighted interval score according to the US COVID-19 Forecast Hub. The usability of these projects improves a lot, if there are consistent with each other.*

Reply: 1–5: We are happy to hear that our revised version addressed your concerns.

6: *As the last suggestion, I would provide the thoughts about which outputs of models (and how) can be useful for decision making. You address the need to discuss ICU usage in the response to the referees' comments. The importance of such outcomes increases at the moment. If you could integrate these thoughts in the publication (even if I understand, that it is not the main goal), this would increase the benefit.*

Reply: In the discussion we have now added a remark on an ongoing debate concerning the practical usefulness of different indicators for the steering of the pandemic. For a long time, seven day incidences have been the main metric used in decisions on intervention measures in Germany. However, there are numerous claims that hospitalization numbers would be more appropriate to this end, as they depend less on testing strategies (even though they are more lagged with respect to the epidemic dynamics). This debate of course translates to the usefulness of forecasts. In particular in a post-vaccination setting, case counts may become a less and less informative quantity to judge the expected healthcare burden. An extension to hospitalization forecasts is currently on the way in the new European COVID-19 Forecast Hub, into which the German and Polish Hub has been largely integrated.